# Contrasting genomic epidemiology between sympatric *Plasmodium falciparum* and *Plasmodium vivax* populations

Philipp Schwabl [1,2,12], Flavia Camponovo[3,4,5,12], Collette Clementson[6,12], Angela M. Early[2], Margaret Laws[1,2], David A. Forero-Peña[7], Oscar Noya[8,9], María Eugenia Grillet[10], Mathieu Vanhove[1,2], Frank Anthony[6], Kashana James[6], Narine Singh[6], Horace Cox[6,11], Reza Niles-Robin [6], Caroline O. Buckee [3] & Daniel E. Neafsey [1,2] ✉

The malaria parasites *Plasmodium falciparum* and *Plasmodium vivax* differ in key biological processes and associated clinical effects, but consequences on population-level transmission dynamics are difficult to predict. This co-endemic malaria study from Guyana details important epidemiological contrasts between the species by coupling population genomics (1396 spatio-temporally matched parasite genomes, primarily from 2020–21) with sociodemographic analysis (nationwide patient census from 2019). We describe how *P. falciparum* forms large, interrelated subpopulations that sporadically expand but generally exhibit restrained dispersal, whereby spatial distance and patient travel statistics predict parasite identity-by-descent (IBD). Case bias towards working-age adults is also strongly pronounced. *P. vivax* exhibits 46% higher average nucleotide diversity (π) and 6.5x lower average IBD. It occupies a wider geographic range, without evidence for outbreak-like expansions, only microgeographic patterns of isolation-by-distance, and weaker case bias towards adults. Possible latency-relapse effects also manifest in various analyses. For example, 11.0% of patients diagnosed with *P. vivax* in Greater Georgetown report no recent travel to endemic zones, and *P. vivax* clones recur in 11 of 46 patients incidentally sampled twice during the study. Polyclonality rate is also 2.1x higher than in *P. falciparum*, does not trend positively with estimated incidence, and correlates uniquely to selected demographics. We discuss possible underlying mechanisms and implications for malaria control.

Malaria caused by the parasite *Plasmodium falciparum* is responsible for more than half a million deaths each year, predominantly in Sub-Saharan Africa and in children under five[1]. Malaria by *Plasmodium vivax* causes less acute mortality but is more widely distributed at the global scale[2,3] and likewise causes severe morbidity and socioeconomic impact[1]. Progress towards malaria elimination is currently stalling in many endemic regions[4], and divergent parasite species' responses to intervention play an important role[5–9]. *P. vivax* in particular often shows stable or increasing incidence during periods in which co-endemic *P. falciparum* is successfully being reduced[4]. It is therefore important to better understand how epidemiological

differences between the species affect the impact of control strategies and whether species-tailored approaches are warranted in co-endemic settings.

*P. falciparum* and *P. vivax* are not closely related (current theory suggests 30–50 million years of independent evolution[10,11]) and differ in key developmental processes and associated clinical effects[4]. For both species, human infection begins when sporozoites enter the skin and vasculature while an infected anopheline mosquito is probing for blood. The parasite first infects the liver and later transitions to blood-stage infection where repeated erythrocyte invasion, intra-erythrocytic replication, and erythrocyte rupture are associated with febrile and paroxysmal disease. A subset of blood-stage parasites commit to onward transmission by differentiating into the sexual, mosquito-infective gametocyte form. Meiosis occurs after transmission to the mosquito via blood meal and involves outcrossing if the blood meal contains genetically distinct gametocytes (i.e., if the human infection source was polyclonal). The parasite forms oocysts under the basal lamina of the mosquito midgut and these later release sporozoites which invade the mosquito salivary glands[12].

*P. vivax* behaves distinctly within the human stages of this infection cycle in multiple key aspects. First, not all *P. vivax* parasites that successfully reach the liver immediately begin to produce merozoites via schizogony as is the case for all liver-stage *P. falciparum* parasites. Rather, a subset of liver-stage *P. vivax* parasites form a dormant hypnozoite reservoir which can initiate additional blood-stage infections weeks to months after initial infection[13]. Clearing this relapsing hypnozoite reservoir from the liver is hindered by incomplete drug efficacy[14], toxicity[15], and adherence concerns[16]. In the blood stage, *P. vivax* invades a smaller subset of erythrocytes[17] and thus more frequently creates submicroscopic infections, especially in adults[18]. Furthermore, while initial blood stage onset is similar between species (12–14 days[12]), the *P. vivax* blood stage generates mature gametocytes ~11 days earlier than that of *P. falciparum*[12]. This increases the likelihood of onward transmission before antimalarial treatment occurs[19,20]. Several experimental infection studies have additionally suggested that the minimum gametocyte density required to infect susceptible mosquitoes is intrinsically lower in *P. vivax* than in *P. falciparum*[19,21–25].

Taken together, the above properties are thought to make *P. vivax* more resilient to clinical interventions targeting symptomatic and often pediatric infection, a main focus of *P. falciparum* control. Vector control measures are also thought to have weaker or delayed effects on *P. vivax* due to post-intervention relapse from undetected or unsuccessfully cleared hypnozoite reservoirs as well as higher probability to achieve onward transmission when mosquito abundance is low. Furthermore, long-lasting hypnozoite reservoirs and more transmissible asymptomatic blood-stage infections may make *P. vivax* more effective at long-distance dispersal and outcrossing with unrelated strains. This may also include greater potential for allochthonous transmission where malaria was previously absent or cleared[26–28].

The epidemiological effects of these biological differences between *P. falciparum* and *P. vivax* have been difficult to quantify because synchronized comparative study designs that avoid confounding by methodological and spatiotemporal factors are challenging to establish[29]. Study systems representing true species sympatry are also rare, as *P. falciparum* and *P. vivax* often occupy distinct, only partially overlapping socio-geographic partitions within areas more broadly considered co-endemic[2,3].

This study uniquely couples a spatiotemporally matched genomic sampling scheme with epidemiological analysis of disaggregated (patient-level) malaria case records to discern co-endemic *P. falciparum* and *P. vivax* transmission dynamics. We focus on Guyana, where both species contribute substantially to the national case count (32.3% *P. falciparum*, 67.4% *P. vivax*, and <1% *P. malariae* in 2019) and where true sympatry is geographically widespread. Malaria in Guyana also involves a diverse mix of ethnicities (e.g., strong representation from Afroguyanese, East Indian, and Amerindian groups) and dynamic population mobility patterns relating to the mining field. Next to important distinctions involving geographic and host demographic variables, we demonstrate clear species contrasts in strain ancestry, persistence, dispersal, and infection complexity within this multi-faceted transmission context. We discuss possible biological drivers behind divergent species epidemiologies and identify various implications for disease control.

## Results

### General spatiotemporal species prevalence

We first conducted a descriptive summary of country-wide passive case detection records made available by the Guyana Ministry of Health (GMOH) for 2006 to 2019. Rolling monthly counts demonstrate well-balanced case burden between species from 2006 to 2010 followed by higher *P. falciparum* counts from 2010 to 2013 (Fig. 1a), a period during which spiking gold prices and La Niña climate conditions are thought to have fueled strong malaria peaks[30]. *P. vivax* cases began to exceed those of *P. falciparum* in mid-2013 and remained over-represented through 2019 (end of dataset). Given the availability of individual-level case metadata curated by the GMOH for 2019, we focused on 2019 records for most subsequent epidemiological analyses. These records include 17,710 single-species cases (67.43% *P. vivax*, 32.35% *P. falciparum*, and 0.22% *P. malariae*) and 1001 mixed-species cases containing both *P. vivax* and *P. falciparum* (Supplementary Table 1). Both species show transmission throughout the year and elevated incidence between April and August (Supplementary Fig. 1).

### Species intricacies of a wide co-endemic distribution

2019 case records show the extent of *P. falciparum*−*P. vivax* geographic and demographic co-endemicity in Guyana (Fig. 1b–d). Regions I, VII, and VIII account for 94.6% of *P. falciparum* cases and 89.3% of *P. vivax* cases, and regional species proportions remain stable across East Indian, Amerindian and Mixed/Other ethnicity groups. While *P. vivax* consistently predominates in these groups, case majority flips to *P. falciparum* in the Afroguyanese (Chi-squared tests, $\chi^2 = 36.13$ (I), 269.63 (VII), 125.51 (VIII), Bonferroni-adjusted (bf-adj.) $p < 0.001$ (all regions), Fig. 1b). This result is consistent with lower *P. vivax* infection risk in Duffy-negative blood types[31]. We further classified infection localities into 27 epidemiological zones (average 3,631 km² each) defined using river-road transport networks and mobility friction rasters (Fig. 1 map, Supplementary Fig. 2). Again, species sympatry is consistently maintained in all zones within Regions I, VII, and VIII. The relative proportion of *P. vivax* cases is generally higher outside of this main malaria range, especially to the south in Region IX (e.g., Upper Rupununi and Greater Lethem). Patients aged 15–54 clearly predominate in both species but are more common in *P. falciparum* (84.5%) than in *P. vivax* (79.2%−Chi-squared test, $\chi^2 = 82.89$, $p < 0.001$, Fig. 1d). The proportion of cases representing *P. falciparum* (Fig. 1e, Supplementary Fig. 3) also increases markedly with age in male patients (Kruskal-Wallis test, $H = 23.64$, $p < 0.001$−see post-hoc significance in Fig. 1e legend). Working-age (≥15 years) to child (<15 years) case ratios are less markedly elevated in Amerindians than in non-Amerindian ethnicities (3.8 to 1 vs. 13.5 to 1, respectively, for *P. falciparum* (Chi-squared test, $\chi^2 = 273.84$, $p < 0.001$) and 2.6 to 1 vs. 7.4 to 1, respectively, for *P. vivax* (Chi-squared test, $\chi^2 = 515.98$, $p < 0.001$), Supplementary Fig. 4). Elevated case bias towards working-age males, more strongly so in non-Amerindian than in Amerindian ethnicities, is consistent with male-dominated hinterland mining activity driving malaria burden in Guyana[30,32] (see mineral resource map in Supplementary Fig. 5), especially in non-resident (e.g., 'coast lander'[33]) mining groups. Higher working-age to

child case ratio for *P. falciparum* than for *P. vivax* is also consistent with the idea that transmission in non-mining communities is less common in *P. falciparum* than in *P. vivax*.

## Deep genomic profiling of each species across 2020 and 2021

To search for epidemiological features that are not observable from traditional symptomatic case analysis, we sequenced 708 *P. falciparum* and 762 *P. vivax* genomes from patient blood spots collected passively by the GMOH. Collection followed informed consent using a study protocol approved by ethical committees of Harvard University and the government of Guyana. Sequencing was successful for 666 *P. falciparum* and 705 *P. vivax* genomes (Supplementary Data 1) continuously spanning January 2020 to June 2021 (Fig. 2). Of these 1371 samples, 1216 (89%) are coupled with patient travel history records used to infer geographic infection source (representing Regions I, VII, and VIII in 1156 (95.1%) samples). We additionally sequenced 13 *P. falciparum* (7 successful) and 22 *P. vivax* samples (16 successful) from 2019 representing infection sites in Venezuela and analyzed two publicly available Venezuelan *P. falciparum* genomes from 2015/16 (European Nucleotide Archive (https://www.ebi.ac.uk) accessions ERR1818176 and ERR2496572). Mean read-depth (MRD) variation among successfully sequenced *P. falciparum* (median MRD = 58.7) and

*P. vivax* samples (median MRD = 28.7) is shown in Supplementary Fig. 6.

## Discrepant genomic diversity and relatedness between species

Genomic analyses focused on 46,067 and 266,141 sites with segregating single-nucleotide polymorphisms (SNPs) identified in the *P. falciparum* and *P. vivax* sample sets (respectively) following quality filtration steps. This contrast in the number of segregating sites and in average pairwise nucleotide diversity observed within each sample set ($1.99 \times 10^{-4}$ differences per bp in *P. falciparum* and $2.91 \times 10^{-4}$ differences per bp in *P. vivax*, Supplementary Fig. 7) aligns with previous evidence that genetic diversity is higher in *P. vivax* at the global scale[34].

To further characterize diversity within each sample set, we applied the hidden state model hmmIBD[35] to estimate parasite relatedness due to recent common ancestry (identity-by-descent; IBD) for 598 *P. falciparum* and 548 *P. vivax* samples identified as single-strain (monoclonal) infections using a Bayesian method called THEREALMcCOIL[36]. The relatedness distributions generated for infections from Guyana (Fig. 3a) clearly differ between the two species (Kolmogorov-Smirnov test, *D* = 0.99, *p* < 0.001). Pairwise relatedness values in *P. vivax* exhibit a mean of 0.048, with very little standard deviation (sd = 0.032) and minimal skew or outlier occurrence. These

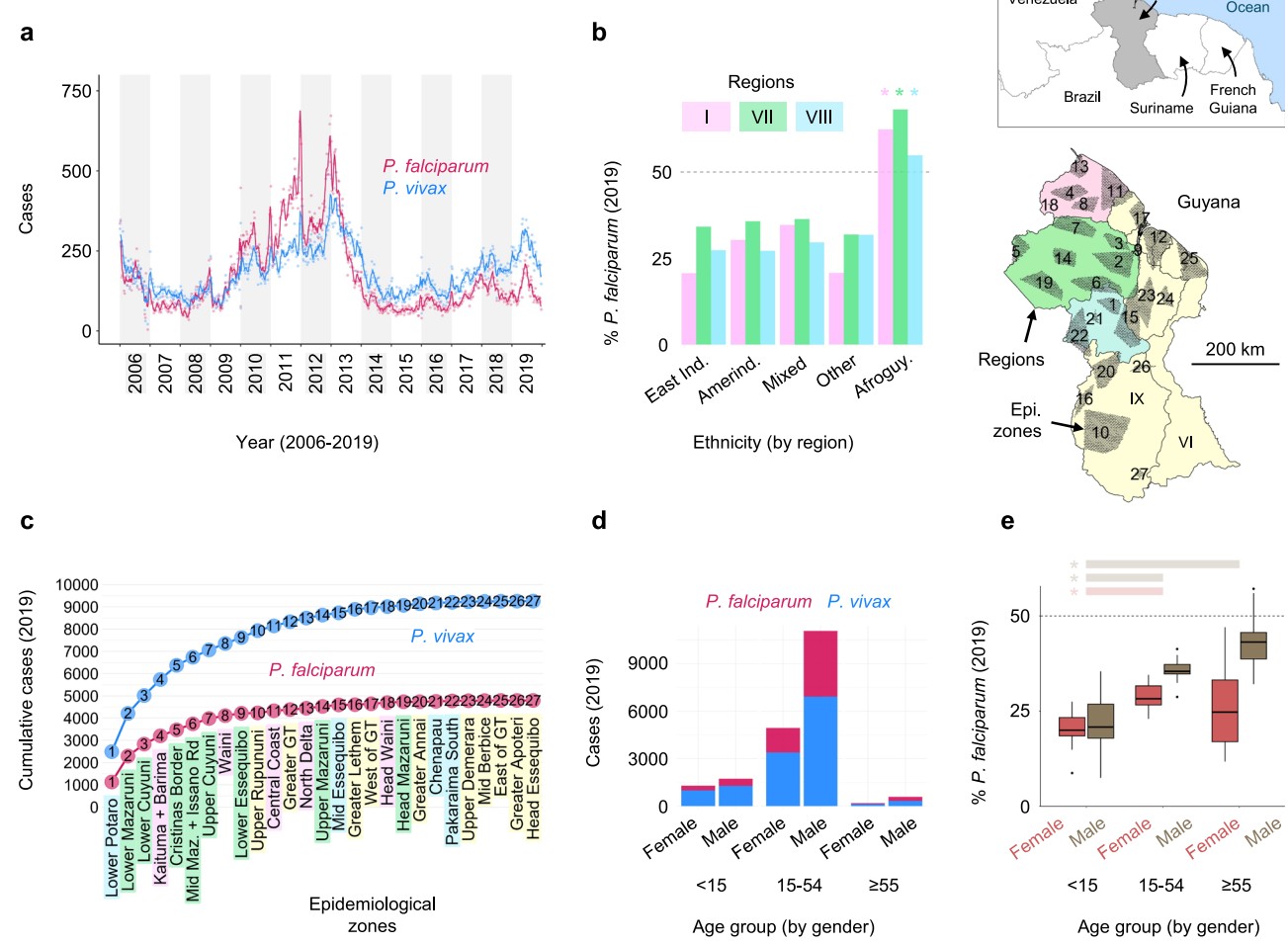

**Fig. 1 | Spatiotemporal and demographic patterns of *P. falciparum* and *P. vivax* in Guyana. a** Weekly reported cases (points) and 30-day rolling average (line) between 2006 and 2019. **b** Percent of cases representing *P. falciparum* (% Pf) by region and ethnicity in 2019. Asterisks indicate significant differences in % Pf between Afroguyanese and non-Afroguyanese cases via Chi-squared test (Bonferroni-adjusted (bf-adj.) *p* = 5.5 × 10⁻⁹ in Region I, bf-adj. *p* = 4.1 × 10⁻⁶⁰ in Region VII, and bf-adj. *p* = 1.2 × 10⁻²⁸ in Region VIII). **c** Cumulative case count across

epidemiological zones in 2019. Zones on the x-axis are ordered by total case count (descending from left to right). **d** Case counts by gender and age in 2019. **e** Boxplots summarize monthly variation (median and quartiles) in % Pf by gender and age in 2019. Asterisks indicate significant differences for age bins <15 vs. 15−54 (Benjamini-Hochberg-adjusted (bh-adj.) *p* = 0.004 in both males and females) and <15 vs. ≥55 (bh-adj. *p* = 4.6 × 10⁻⁶ in males) via two-sided Dunn test.

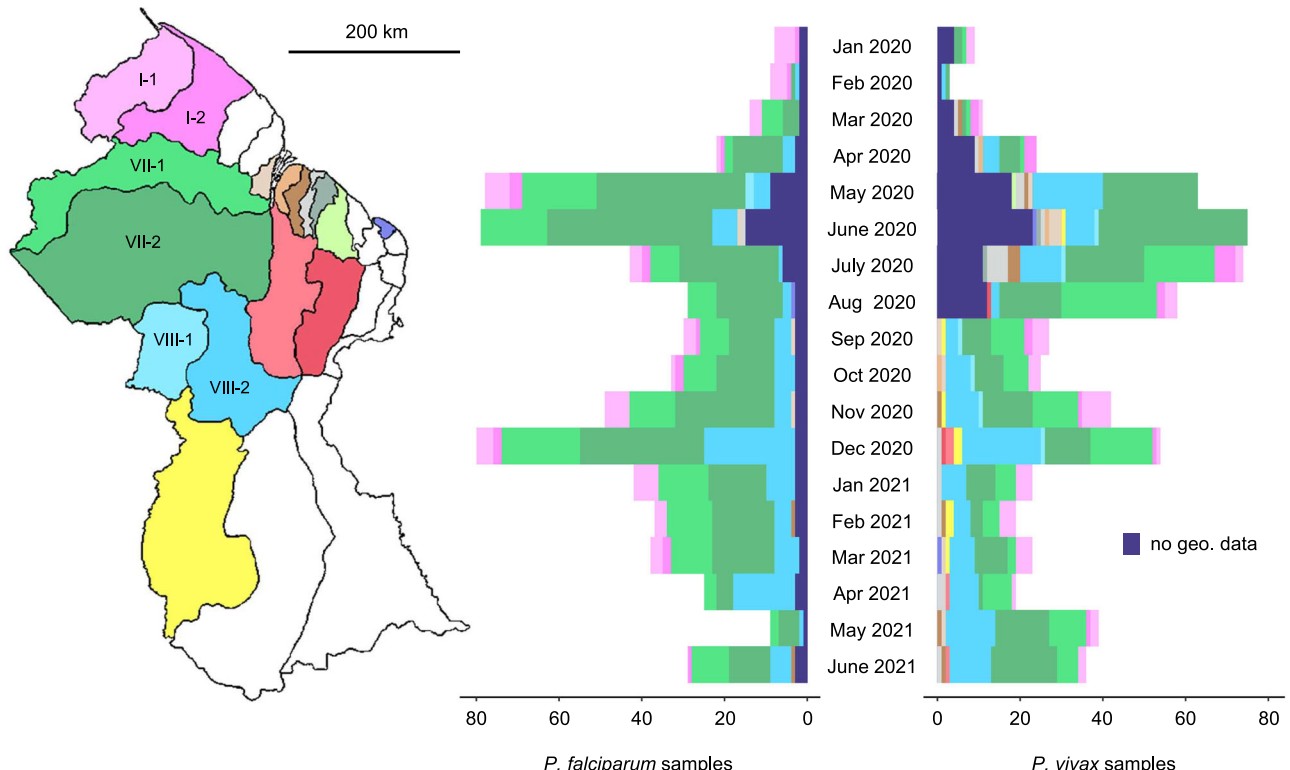

**Fig. 2 | Spatiotemporally matched *P. falciparum* and *P. vivax* sampling in Guyana.** Colors in the map and histograms represent the neighborhood councils (NDCs) to which infections from 2020–21 were attributed based on patient response about location of stay 14 days prior to diagnosis. NDCs within Regions I, VII, and VIII are labeled in the map.

results are indicative of a diverse, frequently outcrossing population without substantial sub-clustering into groups of highly related individuals. In contrast, most pairwise comparisons in *P. falciparum* exhibit greater than half-sibling-level relatedness (mean = 0.310 ± 0.124 sd). The broader, right-tailed relatedness distribution also includes a peak near 1 representing the presence of identical clones as well as values between 0.5 and 0.75 representing the presence of various inbred sibling-level relationships. A network analysis identified 79 clonal groups (>0.90 IBD) among all *P. falciparum* samples, represented by 2–39 members each (Supplementary Fig. 8). Clonal group membership detection was often spatiotemporally aggregated (e.g., >50% of group members detected within 4 months per epidemiological zone, Supplementary Fig. 9), consistent with outbreak-like patterns of prevalence[37]. The largest clonal groups spread across multiple geographic areas (Supplementary Fig. 10a), and most appear highly interrelated (e.g., 50 groups collapse into a single network when re-clustering samples at >0.60 IBD, Supplementary Fig. 8). Four groups noted with asterisks in Supplementary Fig. 8 and mapped in Supplementary Fig. 10b appear slightly divergent (1.7 to 3.3 sd below average between-group IBD). Other clonal patterns mapped in Supplementary Fig. 10b include the clustering of groups #7, #9, and #10 around Port Kaituma and Mabaruma (Region I), groups #18, #24, and #49 along the Mazaruni and Puruni Rivers (Region VII), and groups #27, #32, and #52 around Mahdia and the lower Potaro River (Region 8). Two divergent clonal pairs (#20 and #65) also have partial membership in Venezuela (star symbols in Supplementary Fig. 10b) and likely represent transmission of imported parasite lineages (Supplementary Text 1, Supplementary Fig. 11). Group #20 additionally provides evidence for longer-term persistence of *P. falciparum* clones (detection in 2015 and 2020).

Clonal network analysis results for *P. vivax* are very different. We observe only 29 clonal groups, of which 23 contain 2 members each (i.e., are clonal pairs) and 6 contain 3 members each. Post-hoc review of

patient-specific case codes (blinded to non-GMOH authors) also clarifies that 11 of 23 clonal pairs represent repeated sampling from the same patient (median time between 1st and 2nd visit = 114 days, minimum = 40 days, maximum = 229 days). These likely represent relapses registered as new infections prior to genetic analysis. No close relatedness occurs among any clonal groups (the same number of groups created via clustering at >0.90 IBD remains when re-clustering at >0.30 IBD) or between Guyanese and Venezuelan sample sets (Supplementary Fig. 8, Supplementary Text 1). Furthermore, clonal persistence in *P. vivax* (median = 79 days, maximum = 243 days) appears more temporally limited (Wilcoxon test, W = 55078, p = 0.003) than in *P. falciparum* (median = 114 days, maximum = 500 days, excluding imported lineages (groups #20 and #65), Supplementary Fig. 12).

### Regional isolation-by-distance in *P. falciparum* vs. micrographic structure in *P. vivax*

We next evaluated whether parasite genetic relatedness declines with spatial distance (Fig. 3b). The presence of 'isolation-by-distance' is a common indicator of population structure due to spatially limited dispersal, understanding of which is key to the success of control objectives such as limiting the spread of drug resistance[38,39]. We applied sliding windows of spatial distance, quantifying the fraction of pairwise comparisons in each 30 km window that showed >0.50 IBD (for analysis using mean IBD and other dichotomization thresholds see Supplementary Fig. 13). Results for *P. falciparum* demonstrate a linear decrease in the frequency of >0.50 IBD as spatial distance increases between sample pairs (Pearson's r = 0.89, p < 0.001). Exceptions occur at window starts near 100 km and 200 km where the frequency of close relatives briefly rebounds to levels observed at shorter distance classes. These upticks may partially reflect accumulation of relatedness between popular mining sites or logistics hubs (Supplementary Text 2, Supplementary Fig. 14). In *P. vivax*, sample pairs with >0.50 IBD appear very rare (n = 87) but also clearly overrepresented in the first spatial

**a**

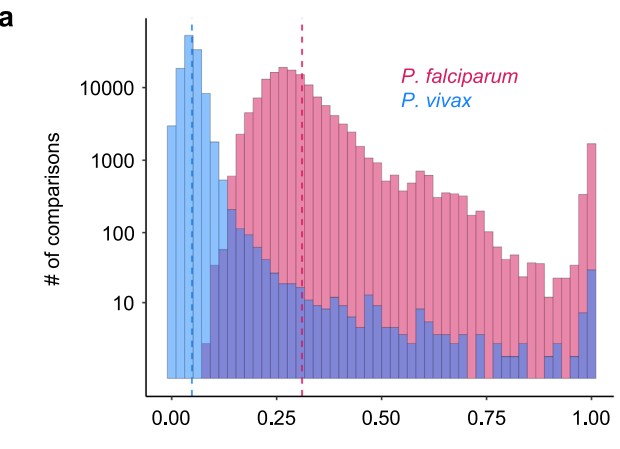

**b**

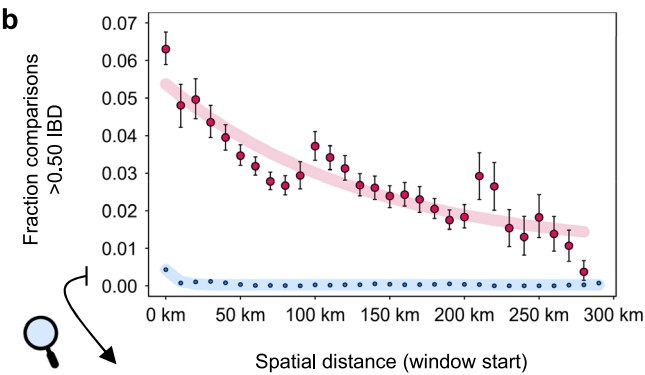

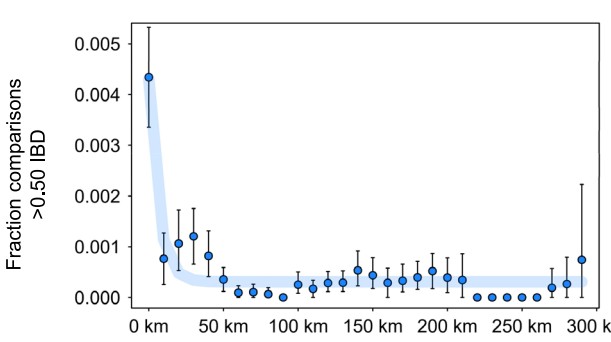

**Fig. 3 | Pairwise IBD and relationships to spatial distance in *P. falciparum* and *P. vivax* in Guyana. a** Histogram of pairwise IBD in 2020–21. The y-axis is log10-scaled to help display less frequent close-relative observations. Dashed lines indicate species means. **b** Relationship of spatial distance between inferred infection sites and the frequency of >0.50 IBD in 2020–21. Points represent observations from windowed calculation using step size = 10 km starting at 0–30 km. Error bars represent 90% confidence intervals from bootstrapping 1000x. The model $y = a*e^{-b*x} + c$ was used to fit regression lines of linear and logarithmic shape. Distance analyses use 359 *P. falciparum* samples and 474 *P. vivax* samples (following an exclusion of *P. falciparum* samples belonging to clonal groups of ≥10 membership size).

distance class (0–30 km, Fig. 3). A continued decline in >0.50 IBD frequency with spatial distance is however not apparent. To help assess whether the spatial distance across which negative correlation between relatedness and spatial distance persists differs between species, we additionally quantified 99th percentile (p99) IBD, an alternative definition of close ancestry that normalizes statistical power in comparative Mantel tests (Supplementary Fig. 15, top). Correlogram results for *P. falciparum* indicated significant correlation between

genetic distance (1–p99 IBD frequency) and spatial distance for 0–30 km (Mantel $r = 0.49$, Holm-adjusted (hm-adj.) $p = 0.001$) as well as for 30–60 km distance classes (Mantel $r = 0.19$, hm-adj. $p = 0.017$). In *P. vivax*, correlation between 1–p99 IBD frequency and spatial distance loses statistical significance at 30–60 km (Mantel $r = 0.12$, hm-adj. $p = 0.089$). Shorter isolation-by-distance signal relative to *P. falciparum* also occurred when using the absolute >0.50 IBD metric to define close ancestry (Supplementary Fig. 15, bottom).

### Genetic vs. non-genetic markers accord better on *P. falciparum* than on *P. vivax* dispersal

We next visualized >0.50 IBD frequency within and between epidemiological zones (Fig. 4) and examined correlations to patient travel patterns inferred from the 2019 epidemiological database. These analyses aimed to identify cases of similar, discordant, or complementary inference on parasite dispersal between data types and to understand the extent to which predictive power differs between malaria species.

Corollary to the clear elevation in >0.50 IBD frequency at 0–30 km (Fig. 3) and significant isolation-by-distance at 0–30 km (Supplementary Fig. 15) in both *P. falciparum* and *P. vivax*, both also generally showed highest frequencies of elevated relatedness within (i.e., not between) epidemiological zones (Fig. 4). This observation from the genomic data matches the observation from the epidemiological database that the majority of symptomatic malaria cases involve non-mobile, 'local' infections (black pie slices, Supplementary Fig. 16), i.e., patients reporting absence of travel 14 days prior to diagnosis. The epidemiological data also suggests that the proportion of local to nonlocal infections (i.e., cases in which patients reported having stayed in a distinct epidemiological zone 14 days prior to diagnosis) is higher for *P. vivax* than for *P. falciparum* in 16 of 23 analyzed epidemiological zones (Supplementary Fig. 17). These 16 zones include non-endemic coastal areas such as Greater Georgetown, where 178 of 1,611 (11.0%) *P. vivax* patients (vs. 45 of 867 (5.2%) *P. falciparum* patients) reported absence of prior travel (Chi-squared test, $\chi^2 = 22.915$, $p < 0.001$). Elevated pairwise nucleotide diversity (+20.7% π) in *P. vivax* samples attributed to Greater Georgetown (Welch's *t*-test, $t = 4.71$, $p < 0.001$, Supplementary Fig. 7) further supports the hypothesis that relapses representing various different geographic infection sources are diagnosed in this capital district. To statistically compare inference of parasite movement based on genetic vs. epidemiological data types, we correlated the relative frequency of parasite >0.50 IBD and patient travel events ('case flow') among all pairs of endemic epidemiological zones for which patient movement was detected in the epidemiological database (i.e., all arrowed segments in Supplementary Fig. 16). In the case of *P. falciparum*, relative frequencies of >0.50 IBD and case flow were moderately correlated (Pearson's $r = 0.44$, $p = 0.029$) between endemic zones (Fig. 5), with most conspicuous congruence in the context of reduced connectivity between Region I and Regions VII and VIII (e.g., see short spokes of elevated >0.50 IBD frequency and intensified case flow around Kaituma and Barima in Region I, yet limited connectivity farther South in Fig. 4 and Supplementary Fig. 16). Only 0.8% of nonlocal infections sampled in Regions VII and VIII (gray pie slices, Supplementary Fig. 16) represented infection sites from Region I. Case flow to Greater Georgetown was also weaker for infections attributed to Region I (36.4%) than for infections attributed to Region VII (43.4%—Chi-squared test, $\chi^2 = 48.22$, bf-adj. $p < 0.001$) or to Region VIII (74.1%—Chi-squared test, $\chi^2 = 874.35$, bf-adj. $p < 0.001$). These results may in part reflect the absence of efficient travel routes from Region I to Georgetown (East) and to regions farther South. Rivers key to travel in Region I (e.g., the Barima and Barama) lead into a Northwest delta while those key to travel in Regions VII and VIII (e.g., the Cuyuni and Mazaruni) flow East into the Essequibo near Georgetown. A handful of semi-developed roads also parallel and connect areas of the Cuyuni, Mazaruni, and Essequibo in Regions VII and VIII.

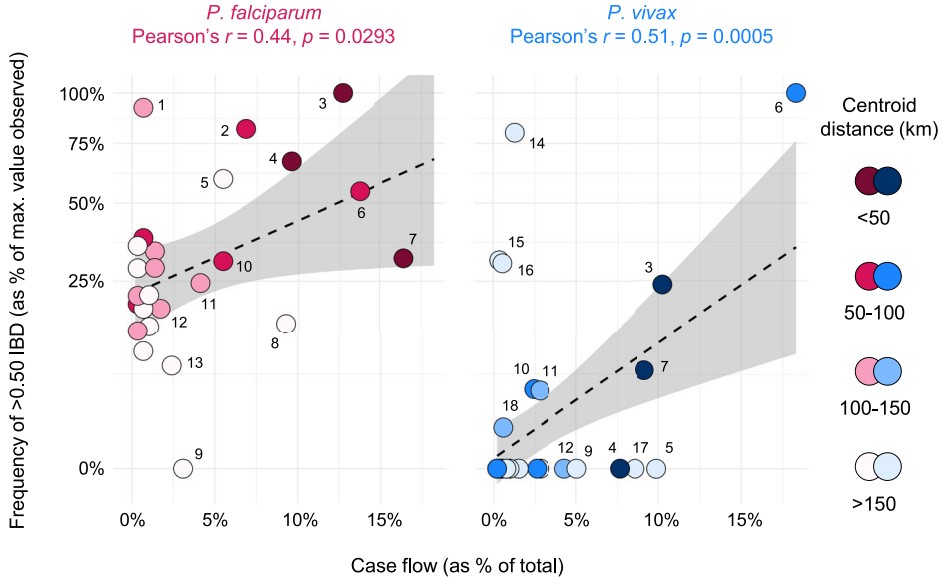

**Fig. 4 | Frequency of >0.50 IBD within and between epidemiological zones in *P. falciparum* and *P. vivax* in Guyana.** Nodes indicate epidemiological zones. Node and segment colors indicate frequency of >0.50 IBD within and between epidemiological zones (respectively) in 2020–21. Node sizes are proportional to genomic sampling size. Between-zone comparisons represented by ≤ 50 comparisons are excluded.

**Fig. 5 | Relationship between patient case flow and the relative frequency of >0.50 IBD in *P. falciparum* and *P. vivax* in Guyana.** Genetic data (*y*-axis) represents 2020–21 and epidemiological (case flow) data (*x*-axis) represents 2019. Point color indicates the spatial distance separating the zones being compared. Comparisons represented by ≤ 50 comparisons are excluded. Gray shading indicates 95% confidence intervals predicted by linear regression (dashed line). Labeled points represent observed values for (1) Central Coast (CC) vs. Lower Cuyuni (LC), (2) Head Waini (HW) vs. Kaituma and Barima (KB), (3) Lower Potaro (LP) vs. Mid Essequibo, (4) KB vs. Waini, (5) CC vs. Lower Mazaruni (LM), (6) KB vs. North Delta (ND), (7) LC vs. LM, (8) Head Mazaruni vs. LM, (9) Cristinas Border vs. LM, (10) LM vs. Mid Mazaruni and Issano Rd, (11) LM vs. Upper Cuyuni, (12) CC vs. KB, (13) LM vs. ND, (14) East of Georgetown vs. LP, (15) LM vs. Upper Rupununi (UR), (16) LP vs. UR, (17) Greater Lethem vs. LM, and (18) LM vs. LP. Please note that significant correlation is lost from the *P. vivax* regression when outlier point #6 is omitted (Pearson's *r* = 0.07, two-sided *p* = 0.675). Significance remains when omitting this point from the *P. falciparum* regression (Pearson's *r* = 0.41, two-sided *p* = 0.045).

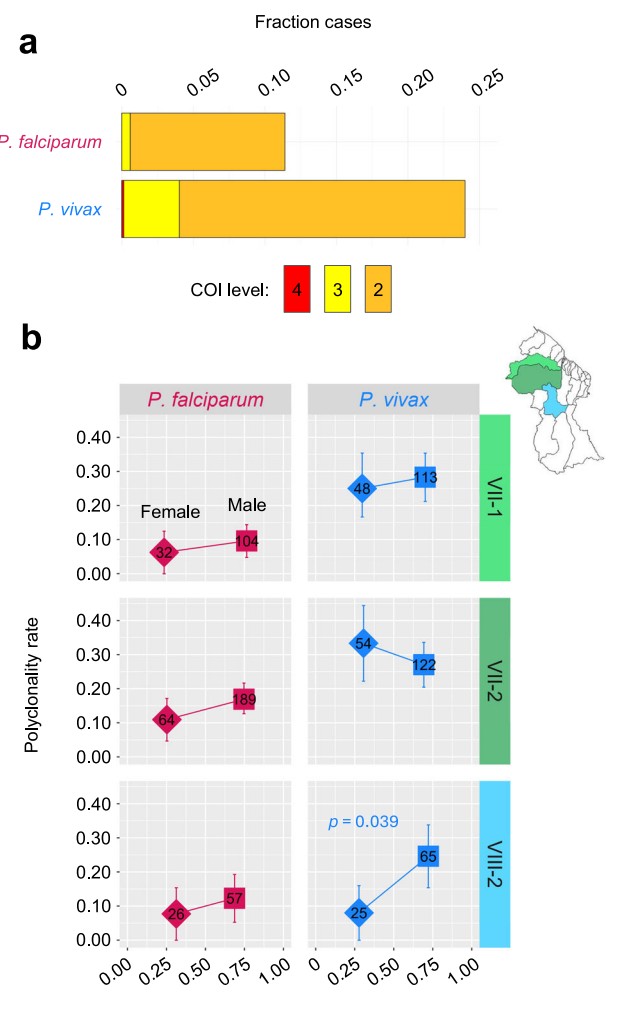

**Fig. 6 | Polyclonality rates and relationships to patient gender in *P. falciparum* and *P. vivax* in Guyana. a** Complexity of infection (COI) values in 2020–21, population-wide. **b** Polyclonality rate (2020–21) vs. patient gender fractions in neighborhood district councils (NDCs) represented by ≥ 20 genomic samples (see sample size annotations within plot). Gender fractions likewise represent 2020–21 (patient metadata from the genomic data set). Square and diamond symbols represent observed values for male and female patients, respectively, and error bars represent 90% confidence intervals from bootstrapping 1000x. Polyclonality rate comparison between male and female patients (two-proportions *z*-test) showed significance only for *P. vivax* in NDC VIII-2 (one-sided *p*-value = 0.039, without correction for multiple comparisons).

For *P. vivax*, the frequency of >0.50 IBD also correlated significantly with case flow between endemic zones (Pearson's *r* = 0.51, *p* < 0.001, Fig. 5). Unlike in *P. falciparum*, however, significance appears outlier-dependent (see Fig. 5 legend) and is lost when broadening the classification of elevated relatedness to p99 IBD (Supplementary Fig. 18). This observation helps clarify that the significant relationships detected when regressing >0.50 IBD frequency on spatial distance (Fig. 3b) and case flow (Fig. 5) do not mean that the *P. vivax* population is predictably structured as a whole. Results are instead consistent with a dispersive, frequently outcrossing *P. vivax* population in which pairwise relatedness is swiftly erased and local genotype associations are rarely established. In this scenario, the ephemeral presence of intact clones or sibling-level relatives creates micrographic spatio-temporal autocorrelations that are not representative of broader population dynamics.

## Higher outcrossing potential in *P vivax*–do relapse and risk carryover play a role?

We further assessed indications from relatedness analyses that outcrossing rate is elevated in *P. vivax* by examining polyclonality rate (i.e., the fraction of samples containing multiple distinct strains) and its variation with respect to geographic and demographic variables. Polyclonality rate is closely related to outcrossing rate because parasite sexual recombination in the mosquito after blood meal only results in outcrossing if multiple distinct strains are ingested simultaneously from the human host.

We observed a polyclonality rate of 24.0% in *P. vivax* vs. 11.4% in *P. falciparum* (Chi-squared test, $\chi^2$ = 36.72, *p* < 0.001, Fig. 6a). Elevated *P. vivax* outcrossing rate implied by this 2.1x polyclonality rate differential (which is likely a conservative estimate due to lower *P. vivax* read-depths–Supplementary Fig. 19) helps explain the aforementioned paucity in clonal *P. vivax* relationships (Fig. 3a) and their limited persistence over time (Supplementary Fig. 12). It is also consistent with the observation that linkage disequilibrium between SNPs declines across physical genetic distance more readily in *P. vivax* than in *P. falciparum* (Supplementary Fig. 20).

Polyclonality rate is considered a positive correlate of *P. falciparum* transmission intensity in the absence of high rates of case importation[40]. We assessed this possible relationship in *P. falciparum* and *P. vivax* (in which polyclonality may additionally occur via hypnozoite-based relapse) by plotting polyclonality rates per epidemiological zone against corresponding incidence estimates (Supplementary Fig. 21). Interestingly, *P. vivax* polyclonality rates remain consistently inflated, regardless of incidence estimate. Polyclonality rate and incidence estimates also trend in opposite directions for *P. falciparum* vs. *P. vivax*, though neither achieves statistical significance. Polyclonality rate was highest in *P. vivax* infections from the Lower Essequibo zone, a key area of travel convergence from the hinterlands of Region VII but one in which malaria risk is not considered high. These results may indicate that *P. vivax* samples from areas of lower endemicity are more likely to represent imported cases (possibly in the form of relapse) that involve multiple accumulated strains.

To further understand polyclonality rate relationships to transmission risk, we compared polyclonality rates in 15–54 year-old males and females within the three neighborhood councils (NDCs) for which ≥20 genomic samples per gender were obtained (Fig. 6b). Males represented the majority of epidemiological case records in the three NDCs, suggesting that transmission risk is generally higher in male-associated occupations. Polyclonality rate in *P. falciparum* thus followed expectations in that observed values appeared consistently higher in males than in females. Polyclonality rates in *P. vivax* were generally more variable (see confidence intervals, especially in females (Fig. 6b)), and were higher in females than in males in NDC VII-2. Interestingly, this trend was influenced by polyclonality rate enrichment in females among patients reporting Venezuelan nationality (Chi-squared test, $\chi^2$ = 4.33, *p* = 0.037, before adjusting for multiple comparisons–Supplementary Fig. 22). This enrichment may occur because infection risk is enhanced in female-biased occupations that accompany the active mining sector (e.g., in rest/supply areas where frequent contact among gametocyte-carrying male miners, mosquitoes, and female non-miners occurs). Enhanced *P. vivax* transmissibility, e.g., due to presymptomatic gametocytemia, may heighten this risk.

## Discussion

In this study we coupled comparative malaria parasite population genomics with passive case record metadata analyses to compare co-endemic *P. falciparum* and *P. vivax* epidemiology in Guyana. We observed several key genomic epidemiological contrasts that indicate the need to consider species-specific approaches to malaria intervention.

Most notably, we observed dramatically different patterns of genomic relatedness between the two species. The *P. falciparum* sample set exhibited a relatively broad pairwise IBD distribution centered near values expected for half-sibling level relatedness, whereas in *P. vivax* this distribution was less variable and clearly shifted towards zero (6.5x lower average IBD). Clonal pairwise relationships (>0.90 IBD) were detected >50x more often in *P. falciparum* and most clonal *P. falciparum* groups exceeded 3 samples in membership size (max. 38). Many of these groups also displayed spatiotemporally aggregated, outbreak-like patterns of detection. In *P. vivax*, clonal groups had no more than 2–3 members and appeared to persist for shorter periods of time; many clonal *P. falciparum* relationships were detected between samples separated by >12 months whereas for *P. vivax* the maximum time between clone detection was 243 days.

This strong contrast in relatedness structure may derive mechanistically from polyclonality rate differences between the species. Polyclonality rate is a key determinant of the extent to which clones and near-relatives can persist in transmission cycles because polyclonality is a prerequisite to effective recombination between strains in the mosquito stage. In this study, *P. vivax* polyclonality was not only 2.1x higher than in *P. falciparum* at the population level but also remained consistently inflated when accounting for estimated incidence in subgeographic comparisons. This result suggests that elevated polyclonality rates in *P. vivax* are not a product of higher infection incidence alone. *P. vivax* relapse from the dormant hypnozoite liver stage likely represents an additional source of polyclonality because relapse can overlap with current infection and can involve the successive or simultaneous activation of distinct strains[41].

Importantly, though humans are the common agent of dispersal throughout Guyana for both parasite species, we observed highly contrasting patterns of spatial genetic structure. The frequency of close *P. falciparum* relatives (>0.50 IBD) decayed with geographic distance, and selected clonal genotypes could be linked to specific geographic subdivisions in Guyana. In *P. vivax*, significant correlation of relatedness with spatial distance did not occur among sample pairs separated by >30 km, and meaningful regional population structure could not be resolved. Limited macrographic structure in *P. vivax* may occur largely because the dormant hypnozoite liver stage unique to this species facilitates long-distance dispersal via relapse in mobile hosts. Various study results point to the influence of relapse on *P. vivax* dispersal in the study region. At health centers in Greater Georgetown, for example, an absence of travel 14 days prior to diagnosis was reported more than twice as frequently by *P. vivax* patients (178 of 1611 (11.0%)) than by *P. falciparum* patients (45 of 867 (5.2%)). Nucleotide diversity was also significantly elevated in *P. vivax* samples representing such non-traveler cases. Given that Greater Georgetown is considered free of malaria transmission, many non-traveler *P. vivax* cases may represent reactivation of dormant parasite diversity that has accumulated in hosts over longer periods of time[41]. We also observed the recurrence of *P. vivax* clones in 11 patients for which blood spots were incidentally sequenced from separate malaria visits. These same-patient sample pairs represented symptomatic malaria episodes separated by a median of 114 days and the prescription of a 3 day Chloroquine + 14 day primaquine treatment course. They may thus not only represent direct evidence of relapse but also suggest efficacy and adherence challenges relating to hypnozoicidal drugs. Together, these observations suggest that differences in underlying parasite biology are contributing to divergent species responses to intervention, including flips from *P. falciparum* to *P. vivax* case majority observed over the last 15 years in various regions nearing elimination goals (e.g., Solomon islands, Myanmar, Cambodia, Lao PDR[1]).

In the case of *P. falciparum* in Guyana, malaria control strategy may benefit from further advancing decentralized case management and differentiating resource allocation based on real-time spatiotemporal surveillance. Similar to regions in Southeast Asia where these approaches have recently reduced *P. falciparum* cases to historic lows[42], our study indicated a *P. falciparum* metapopulation maintained by periodic expansions from high-risk transmission areas (mines in remote riparian forest) where infrastructure is limited and occupational incentives conflict with self-care. In these settings, treatment-seeking is more likely to be delayed (increasing the risk of transmission prior to treatment) and access to regional treatment centers may involve long-distance travel through endemic terrain. It may therefore be beneficial to further decentralize medical capacity from regional treatment centers to localities with strong outbreak signals (e.g., Puruni Landing) or to strategic points along travel routes that may facilitate parasite dispersal (e.g., along the Mazaruni and Cuyuni Rivers, possible conduits for clonal *P. falciparum* dispersal toward Bartica based on this study). Self-administered diagnostic and treatment approaches (e.g., Malakit[43,44]) might additionally help cover remote areas, with monitoring to evaluate the extent to which incorrect self-treatment may negatively impact user health or contribute to resistance emergence.

In the case of *P. vivax* control, our study suggests that similar strategies of decentralized and focally intensified intervention will be less efficacious. We do not observe tractable metapopulation structure that would help prioritize intervention foci, and this lack of structure points to the risk of latent dispersal processes undermining geographically differentiated intervention. Results emphasize the importance of radical cure (treatment combining blood schizonticidal and liver hypnozoicidal drugs[45]) to help reduce *P. vivax* spread, but implementing radical cure in Guyana's remote transmission contexts poses significant challenge. Primaquine, currently the only hypnozoicidal drug specified in Guyana's National Treatment Guidelines, must be taken daily for 2 weeks and requires prior glucose-6-phosphate dehydrogenase deficiency assessment of hemolysis risk. Field trials on new implementation strategies (e.g., CUREMA[46]) and alternative hypnozoicidal treatments (e.g., single-dose tafenoquine) tailored to remote mining settings are therefore important. If safety and adherence concerns can be resolved, the concept of universal radical cure (extended hypnozoicidal treatment not only for *P. vivax* but also for *P. falciparum* patients) also deserves consideration. Universal radical cure has been proposed for transmission contexts in which *P. falciparum* patients are likely to carry hypnozoites from earlier *P. vivax* infection[47]. This condition clearly appears relevant to Guyana given its high levels of spatiotemporal co-endemicity and clear risk profile for co-infection (adults of non-African descent involved in the mining sector) described in this study.

Fortifying preventive vector control and risk awareness programs is also important. A recent study suggests that increased risk perception is associated with increased preventive behavior against vector-borne disease in Guyana, although effect size is small (4–5%)[48]. Effect size might be increased if educational programs more strongly convey risk of onward malaria transmission as opposed to risk solely towards personal health. Such altruistic risk perception may be relevant to the more rapid, pre-symptomatic spread of *P. vivax* malaria into local vectors, including upon miner return to non-mining home environments.

This study had several limitations. Importantly, the near-relative detection approach we used to describe parasite population structure focuses on outlier IBD relationships and therefore underuses available sequence content. We also did not attempt IBD estimation involving polyclonal infections[49], which constituted 17.9% of our genomic sample set. Furthermore, genomic sample collection relied on a passive sentinel method which may bias against parasite diversity found in remote, disconnected transmission cycles (e.g., Amerindian settlements) and against low-parasitemia infections. It was also challenging to address the transitory nature of human settlement in the hinterlands of Guyana, where conventional population sizes are not recorded or do not apply. This primarily complicated the estimation of

zonal incidence and its relationship to polyclonality rates. Host movement through undercharted hinterland areas also added uncertainty to geographic infection source estimates.

Despite these limitations, this study clearly exposes the discrepant profiles of co-endemic *P. falciparum* and *P. vivax* malaria epidemiology in a low transmission context and emphasizes the need for distinct approaches to surveillance and intervention. It has been very important to prioritize investment in drugs, vaccines, and monoclonal antibodies for *P. falciparum* as a more frequent cause of acute severe disease, especially in children. Weaker focus on *P. vivax* control and specialized intervention however makes it likely that malaria will persist in many co-endemic regions after *P. falciparum* is eliminated, forestalling the ultimate goal of eliminating this major disease from all regions and demographics.

## Methods

### Epidemiological database and definitions

The epidemiological database analyzed in this study comprised all malaria episodes detected in Guyana via passive surveillance (estimated to represent ~80% of cases[1]) and reported to the National Malaria Program between 2006 and 2019. Malaria episodes were defined using both medical diagnoses and parasitological tests (blood microscopy and rapid diagnostic tests). Patient records were curated by the Vector Control Services (VCS) division of the GMOH and anonymized records of patient age, gender, nationality, and self-reported ethnicity (categorized as 'Afroguyanese', 'Amerindian', 'East Indian', 'Chinese', 'Portuguese', 'European', 'Mixed', or 'Other') were provided to the Harvard T.H. Chan School of Public Health (HSPH) following approval by the Harvard University Area Human Research Protection Program (protocol IRB18-1638) and by ethical committees of the GMOH. Infection localities were designated by asking *Where [the] patient stayed 2 weeks ago* (see survey form in Supplementary Fig. 23). Each voluntary travel history response was classified to one of 460 country-wide localities with known latitude/longitude coordinates. Infection localities were entered as missing (NA) if patients provided no response on prior stay (i.e., infection localities were not equated to diagnosis localities unless patients specified having stayed in the same locality 2 weeks ago). For foreign localities of prior stay (e.g., in neighboring Venezuela), geographic specifics were not retained in the database. The prior stay field simply shows the term 'imported' for these cases, which we re-code as 'foreign' in our analysis. To define epidemiological zones, we mapped malaria survey sites used by the GMOH (https://gazetteer.glsc.gov.gy/gazetteer/#7/3.650/−57.129) onto a custom shape file containing Guyana's primary river and road coordinates and onto a motorized transport resistance raster obtained from https://malariaatlas.org/ (Supplementary Fig. 2). Sites were clustered based on river/road connectivity in the R package RIVERDIST[50] v0.16.3, travel conductance using the R package GDISTANCE[51] v1.6.4, and manual assessment of river/road and resistance layers in QGIS[52] v2.18.4. Transmission intensity (incidence) was estimated for each epidemiological zone using population size projections for 2019 and 2021 by the LandScan Project[53]. LandScan applies dasymetric modeling to disaggregate census counts based on remotely sensed images[53]. To estimate incidence for an epidemiological zone, we first cropped LandScan Global raster data (https://landscan.ornl.gov/) to a circle of 45 km radius around the centroid of all infection localities belonging to that zone. We then divided the case count (i.e., the number of *P. falciparum* and/or *P. vivax* cases representing patients which reported prior stay in a locality belonging to the zone) by the sum of all values projected within the cropped raster.

### Clinical sample material, sequencing, and variant calling

Following informed consent to analyze parasite genetic polymorphism using patient blood, samples were collected from microscopy or RDT-positive patients by spotting ~50–200 μl whole blood onto Whatman FTA filter paper cards. Collection was carried out by VCS between January 2020 and June 2021 at medical facilities in Bartica, Georgetown, Mahdia, Lethem, and Port Kaituma, Guyana. Samples were stored at room temperature using individual desiccant packets before shipment to HSPH. The same anonymized patient metadata variables as described in the previous section (Supplementary Fig. 23) were also provided to HSPH. An additional 23 dried blood spot samples were collected in Venezuela at the Institute of Tropical Medicine, Central University of Venezuela (Caracas, bioethics permit CEC-IMT 12/2013) and Biomedical Research and Therapeutic Vaccines Institute (Ciudad Bolívar, bioethics permit CHURPCBBS-008-2019) in 2019. Metadata for these samples (collection date and reported place of stay (municipality) 2 weeks prior to diagnosis) was likewise de-identified prior to HSPH access (Supplementary Data 1).

We extracted total genomic DNA from dried blood spot samples (~20–35 mm² spotted area punched per sample) using KingFisher Ready DNA Ultra 2.0 Prefilled Plates on the Kingfisher Flex instrument (ThermoFisher Scientific). We subsequently applied whole-genome amplification to DNA extracts, each 50 μl sample reaction consisting of 5 μl 10x phi29 polymerase buffer (NEB B0269S), 0.125 μl (2.5 μg) recombinant albumin (NEB B9200S), 0.5 μl primer mix (10 oligos combined at 250 μM, see below), 5 μl 10 mM dNTPs (Thermo Scientific), 26.375 nuclease-free water, 3 μl (30 units) phi29 DNA polymerase (M0269L), and 10 μl DNA. Reactions were prepared on ice, with components added in the order shown. Primer mixes consisted of 'Pf1-10' for *P. falciparum*[54] and 'Pvset1' for *P. vivax*[55] (set selection based on previous microscopy or RDT-based species assignments). The 3' regions of these primers contain phosphorothioate bonds. Amplifications were generated using step down incubation (35 °C for 5 min, 34 °C for 10 min, 33 °C for 15 min, 32 °C for 20 min, 31 °C for 30 min, 30 °C for 16 h, and 65 °C for 15 min), followed by cooling to 4 °C. We then applied AMPure XP magnetic beads (Beckman Coulter A63881) at room temperature to exchange post-reaction sample buffer to 10 mM Tris-HCl + 0.1 mM EDTA. Final library construction using the NEBNext Ultra II FS DNA Library Prep Kit (NEB E6177) and 2 × 151 bp sequencing on the Illumina NovaSeq 6000 platform was completed at the Broad Institute. We aligned reads to the *P. falciparum* 3D7 and *P. vivax* P01 reference genome assemblies using BWA-MEM[56] v0.7.17-r1188 and called SNPs and INDELs using GATK[57] v3.5-0-g36282e4 'HaplotypeCaller' and 'GenotypeGVCFs' according to best practices defined by the Pf3k consortium (http://www.malariagen.net/data_package/pf3k-5/). The mapping and joint variant call process also included sample accessions ERR1818176 and ERR2496572 representing malaria infections from Venezuela (further geographic specifics unknown). All downstream genetic analyses focused on SNP sites in core regions of the genome[58,59] and >5 bp from any INDEL call. Samples missing genotype calls for >50% SNP sites at ≥2% minor allele frequency (after application of the core region filter and INDEL masking steps) were also excluded from further analysis. Among excluded samples, coverage (% bases represented by >5 reads) averaged 12.9 in *P. falciparum* and 9.9 in *P. vivax*. Following sample exclusions, we removed sites at which >7.5% of monoclonal samples showed multi-allelic ('heterozygous') calls. These filtration steps generated the base SNP call sets for each species. Additional, analysis-specific settings are noted below.

### Parasite genetic analyses

We assessed polyclonality using THEREALMcCOIL[36] v2, a Bayesian Markov chain Monte Carlo approach in which posterior distributions for the population allele frequency of each locus and the COI of each individual are jointly estimated to maximize the likelihood of the data using a Metropolis-Hastings sampling approach. The probability of error (i.e., incorrectly calling a homozygous locus as heterozygous and vice versa) can also be estimated alongside allele frequencies and COI by specifying err_method = 3. We applied the categorical method (i.e., input data = SNP calls classified binarily as homozygous or

heterozygous regardless of the signal intensity of each component allele) with err_method = 3 to bi-allelic SNP sites with ≥10% minor allele frequency and ≥90% call success across samples. We further sub-sampled input to limit computational costs. Subsampling using the '--thin' function in VCFtools[60] v0.1.15 resulted in 337 *P. falciparum* and 350 *P. vivax* input sites. We used the lower bound (quantile = 2.5%) output value as a conservative estimate of sample COI. Estimates were consistent across repeat runs (100% matching calls) and when reducing input sites to *n* = 327 for both species by further filtering out sites with ≥200 mean and ≥150 sd read-depth (99.2% matching calls). Analyses of genetic diversity and relatedness only included monoclonal samples. We estimated nucleotide diversity from the base SNP call sets (previous section) by measuring average nucleotide differences over discrete 100 kbp windows (no overlap) with VCFtools[60] v0.1.15. We measured linkage decay over bi-allelic SNP sites by recoding calls to non-reference allele counts (0, 1, or 2) and computing linkage ($r^2$) in sliding 10 kbp windows (200 bp steps) in PLINK[61] v1.90b1g.

We estimated genetic relatedness between samples based on the concept of IBD. Unlike identity by state (IBS), IBD specifically represents genetic similarity inferred to have been inherited from a common ancestor based on factors affecting linkage likelihood such as population allele frequency, recombination rate, and chromosomal distances between variant sites. IBD generally enhances representation of recent ancestry in obligately sexual species in which divergence history is influenced more strongly by recombination than by the individual accumulation of point mutation events. We ran hmmIBD[35] v2.0.4 in default settings to estimate population allele frequency and infer presence or absence of IBD at all bi-allelic SNPs with ≥2% minor allele frequency and ≥70% call success across samples. We used the fraction of all variant sites inferred as IBD ('fract_sites_IBD') to summarize relatedness for each sample pair.

We used the R package IGRAPH[62] v1.3.5 to generate connected graphs (clusters) in which edges represent sample pairs with relatedness values above a specified threshold. Clusters therefore represent groups of samples (nodes) in which each sample is connected by at least one direct edge to another sample (but simultaneous direct connections to additional cluster members are not required).

We also correlated relatedness values to Euclidean distances calculated from latitude and longitude (WGS 84) coordinates projected onto a common xy plane (EPSG 3786). We used Mantel tests to assess statistical significance between genetic and spatial distance matrices using the R package VEGAN[63] v2.6-4. We applied repeated random draws of 10 samples from each epidemiological zone (drawing until depletion without replacement) to build each distance matrix. We also mapped the frequency of pairwise relatedness between epidemiological zones using shape files from the Database of Global Administrative Boundaries v4.1[64].

Finally, we also assessed the possibility that cases of clonal IBD detection in *P. vivax* represent repeated blood spot sampling events from the same patient (e.g., a first clinical visit due to primary infection and one or more later clinical visits due to relapse or recrudescence by the same parasite genotype). HSPH authors sent a query list of 705 *P. vivax* samples to the GMOH and the GMOH flagged each according to whether it represents a patient which is also represented by another sample in the query list. The list returned by the GMOH indicated 1x (i.e., non-repeat) patient representation for 589 samples, 2x patient representation for 92 samples (necessarily representing 46 patients), and ≥3x patient representation for 22 samples (necessarily representing ≤7 patients). The returned list did not specify which sample sets corresponded to the same patient.

### Reporting summary
Further information on research design is available in the Nature Portfolio Reporting Summary linked to this article.

## Data availability
The sequence data generated by this study has been deposited in the NCBI Sequence Read Archive (https://www.ncbi.nlm.nih.gov/sra) under BioProjects PRJNA809659 and PRJNA1092690. The epidemiological case data (2019) is however not directly available for public use. This reflects a data use agreement with the Guyana Ministry of Health which protects against the potential for subject identification and other privacy/ethics issues. Interested parties may contact the Guyana Ministry of Health (https://www.health.gov.gy/index.php/contact-us) for data access through a data use agreement. All inquiries will be addressed within a reasonable timeframe (3 months or less).

## Code availability
Analysis scripts and files are available at https://doi.org/10.5281/zenodo.13351513.

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

## Acknowledgements

We thank the participants who contributed blood samples to the study, as well as the technicians who collected and processed the samples. This study was supported by the Bill & Melinda Gates Foundation (INV-009416 to D.N. and C.B.). Under the grant conditions of the Foundation, a Creative Commons Attribution 4.0 Generic License has been assigned to the Author Accepted Manuscript version that might arise from this submission. This study was also supported with federal funds from the National Institute of Allergy and Infectious Diseases, National Institutes of Health, Department of Health and Human Services, under Grant Number U19AI110818 to the Broad Institute. Sampling in Venezuela was additionally supported by the Scottish Funding Council Global Challenges Research Fund (GCRF) Small Grants Fund SFC/AN/12/2017 and the GCRF Vector-borne Disease Control Network (EP/T003782/1) led by the Llewellyn Lab (University of Glasgow).

## Author contributions

P.S., F.C., D.N., and C.B. conceived the study. H.C., R.N., D.F., O.N., M.G., and M.L. managed sample collection and logistics. F.A., N.S., and K.J. provided oversight. C.C. curated the epidemiological database. P.S. and D.N. supervised laboratory work. P.S. and F.C. analyzed the data. A.E. and M.V. assisted in analysis and interpretation. P.S. wrote the original manuscript. F.C. and D.N. helped review and edit additional drafts. All authors reviewed and approved the final manuscript.

## Competing interests

The authors declare no competing interests.

## Additional information

[1]Department of Immunology and Infectious Diseases, Harvard T.H. Chan School of Public Health, Boston, MA, USA. [2]Infectious Disease and Microbiome Program, Broad Institute of MIT and Harvard, Cambridge, MA, USA. [3]Center for Communicable Disease Dynamics, Department of Epidemiology, Harvard T.H. Chan School of Public Health, Boston, MA, USA. [4]Department of Epidemiology and Public Health, Swiss Tropical and Public Health Institute, Basel, Switzerland. [5]University of Basel, Basel, Switzerland. [6]National Malaria Program, Ministry of Health, Georgetown, Guyana. [7]Biomedical Research and Therapeutic Vaccines Institute, Ciudad Bolívar, Venezuela. [8]Institute of Tropical Medicine, Faculty of Medicine, Central University of Venezuela, Caracas, Venezuela. [9]Center for Malaria Research, Institute of Higher Studies 'Dr. Arnoldo Gabaldón', Ministry of Popular Power for Health, Caracas, Venezuela. [10]Institute of Zoology and Tropical Ecology, Central University of Venezuela, Caracas, Venezuela. [11]Caribbean Public Health Agency, Port of Spain, Trinidad and Tobago. [12]These authors contributed equally: Philipp Schwabl, Flavia Camponovo, Collette Clementson. ✉e-mail: neafsey@broadinstitute.org

