## [Peer Review File · Nature Communications]

Contrasting genomic epidemiology between sympatric *Plasmodium falciparum* and *Plasmodium vivax* populationsREVIEWER COMMENTS

Reviewer #1 (Remarks to the Author):

NCOMMS-24-21960

As many regions make significant reductions in malaria incidence globally, efforts to reduce the incidence of *P. vivax* have generally proven less successful than those for *P. falciparum*. Various *P. vivax* biological features like relapsing hypnozoites may be driving these challenges, but the epidemiological effects of these differences have been understudied due to a dearth of analyses in spatiotemporally matched genomic samples from sympatric populations. This manuscript contrasted the patterns of genomic relatedness and spatial structure between sympatric populations of two species in Guyana to assess the potential need for species-specific interventions. *P. vivax* isolates were more diverse, exhibited fewer clonal groups, and had shorter isolation-by-distance signals than *P. falciparum*. The frequency of highly related *P. vivax* infections correlated less closely with epidemiologic travel events than with *P. falciparum*. *P. vivax* had a higher polyclonality rate, which is a likely mechanistic contributor to the differences in diversity and relatedness in this species compared to *P. falciparum*. This study makes a meaningful contribution to the field, presenting an analysis of novel sympatric Plasmodium populations, providing important insights into the differences in population and spatial genetic structure, and offering potential implications of these insights for malaria elimination efforts in the country. Study limitations, some moderate and some minor, are noted below.

Lines 178-179: The manuscript notes that data from 250 polyclonal infections were excluded from the analysis; however, it does not provide the number of samples excluded for each species or final sample sizes for both species after removal of polyclonal infections. This information can be calculated using the percent polyclonal isolates noted later in the text or the supplement, but it would be good to have a clearer final sample size for each species reported earlier in the main text. In addition, since polyclonal infections are more prevalent among *P. vivax* infections, it would be good to discuss how exclusion of more *P. vivax* isolates from the analysis impacted downstream inferences. Also, why not try to salvage some of the polyclonal samples that have a predominant clone?

It is not clear why the authors chose to dichotomize IBD levels into >0.5 IBD sharing or not. Figure 3a indicates a wide range of total cumulative IBD sharing values, particularly for *P. falciparum*, that could be used to treat IBD as a quantitative variable, or at least a more granular ordinal variable, rather than dichotomous. A lot of information is lost by treating the data as dichotomous, which could provide better resolution, particularly for *P. vivax*, which had very few infections sharing >0.5 IBD. The authors note in the discussion that examining only outlier IBD relationships is a limitation, but do not provide a justification for why the analysis was performed this way.

The patterns of isolation-by-distance shown for *P. falciparum* shown in Figure 3b are very strange with almost a cyclical pattern in specific distance bins. The authors suggest that this pattern may be driven by clonal outbreaks in commonly visited locations within these distance ranges. It would be helpful for the authors to discuss this observation in more depth. Do the epidemiological data support their hypothesis? Did they conduct a sensitivity analysis, using different sized distance bins, or excluding identical clones, to try to better understand this pattern? Also, were genomes sampled from the same person at different time points excluded from the relatedness analyses, since these are not independent infections?

It would be helpful to know the average read depth of the sequence data from *P. falciparum* and *P. vivax* and how that might have impacted inferences of COI. I anticipate that parasitemias (and thereby read depth) might be greater for *falciparum*, which could make the reported COI differences between the species less pronounced (may more easily identify minor clones in *falciparum* infections than in *vivax* infections).

It would also be helpful for the authors to discuss in the methods how the infection location was designated, i.e., was it designated based on location of diagnosis, or for individuals who traveled, based on travel location?

Figure 5 – The correlation between IBD >0.5 and case flow is higher for *P. vivax* than for *P. falciparum*, but this may be strongly driven by one outlier observation (#6), because overall, the correlation appears to be lower for *P. vivax* than *P. falciparum*.

Other minor points:

Figure 1. Figure legend does not explain the final map on the left of the top row that defines the epidemiological zones. In the text (line 138), figure 1C is referred to and seems to reference the map with epidemiological zones, but 1C is currently the graph of cumulative cases. The numbers of the epidemiological zones are never directly linked with the names used in the x-axis of Figure 1C, making it confusing to determine where on the map the authors are referring to in later discussions of results. Figure 1E: What is the quantitative estimate of significance? In Supp. Fig 12, it is not clear from the legend how “Foreign Import” cases were defined and how that differed from non-local cases.

There are no epidemiological region numbers indicated on Fig 4 or Supp Fig 12, which makes some of the patterns described in the main text difficult to discern, e.g., reduced connectivity between Region I and Regions VII and VIII.

Is there a justification for the use of a 2% MAF cutoff? Many studies use 1% as a MAF cutoff (which is arbitrary), but I wasn't sure if there was a specific rationale here for using a more stringent cutoff, when having more diversity may provide better resolution.

In Supplementary Figure 8, semi-transparent symbols are hard to see, and group IDs are difficult to distinguish by color (e.g., 19, 31, 23, 33).

Supplementary Tables 1 and 2 are flipped from the notations in the text. Table 2 currently corresponds to the 2019 infections, and Table 1 corresponds to the metadata for sequenced isolates.

Reviewer #2 (Remarks to the Author):

Reviewer #3 (Remarks to the Author):

The manuscript by Schwabl et al entitled “Contrasting genomic epidemiology between sympatric *Plasmodium falciparum* and *Plasmodium vivax* populations” uses standard epidemiological data and genomic features to compare the epidemiology of *P. vivax* and *P. falciparum* in Guyana. The study gives a comprehensive and complete comparison of Pf and Pv population genomics, highlighting the main differences found and connecting them with suggestions for control and elimination strategies. The results show how genomic analysis can identify epidemiological features that are not observable from traditional symptomatic case analysis. The results are relevant, robust and well presented. The authors present many possible interpretations for the results, although these are hard to prove.

Major comments:

There is a chronological discrepancy between the standard epi data (mainly 2019) and genomic data (2020 and 2021). Why did authors decide to focus on epi data from 2019 and not from 2020/2021? Figure 1a shows a decline of malaria cases starting in 2019 that may affect the results (ie, relating polyclonality in samples from 2020-21 with incidence data from 2019; Fig 6b).

Most of the analysis is centered on relatedness and polyclonality, which are highly sensitive to how informative of diversity are the used markers (SNPs) for each species. The results of this study point to higher diversity, polyclonality and lower relatedness (IBD) in Pv than in Pf: could this be a consequence of the chosen SNPs (ie, especially diverse for Pv compared to Pf, not necessarily meaning that Pf strains have a higher "real" IBD)? In particular, the analysis is done for 337 SNPs for Pf versus 350 SNPs for Pv: could this imply lower IBD and higher polyclonality statistics from Pv than from Pf just by statistical chance? Also, the filtering of >10% minor allele frequencies could imply different exclusions for the different species, especially if one has a larger presence of low frequency alleles. Can authors think on a way to prove that the comparisons of the study are not biased due to the selection of markers (SNPs)?

Minor comments:

The results sections is accompanied by frequent interpretation of the data (ie, 151-154, 227-228, 266) which may be better allocated in the discussion section. ç

Authors explore the relationship between polyclonally and transmission intensity. Naturally immunity (and therefore intensity of transmission=exposure levels) shape parasite densities, which in turn can affect the ability to detect polyclonal infections (ie, high parasite densities, higher probability of detecting multiple genotypes if present). Do authors think this is something that should be taken into account in the analysis?

Lines 185-189: Why is >0.90 IBD the cutoff to define clonal groups? Why not 0.95 or 0.99, for instance? Is there a biological, statistical or referential reason for this?

Lines 122-123: The authors show the temporal trends of incidence from both species, showing fluctuations between their relative abundance, with some periods dominated by Pv and some other periods by Pf. It would be very interesting to know if the authors have any interpretation of hypothesis for these changes.

In general, but especially in Results: Species intricacies of a wide co-endemic distribution: are the p-values corrected for multiple comparisons?

Line 149-151: The authors mention an elevated case-bias towards non-Amerindian working-age males. Do we have a statistical significance for this statement?

Line 176: I would mention the name of the hidden state model here for clarity.

Line 178-179: I would mention more details of the Bayesian framework used to exclude the 250 samples.

Lines 325-326: is the male representation coming from a possible bias in reporting of care seeking behaviour?

Lines 431-432: regarding limitations, the fact that IBD was conducted only on monoclonal infections could imply some bias in the statistics? This would imply a larger exclusion for Pv than for Pf, for example.

As authors mention, spatiotemporally matched sampling schemes are very rare and challenging to be done. Should this be better recognized in the distribution of authors? There seems to be a tendency toward middle positions from the malaria-endemic authors. While this is not an element that I consider as relevant for the evaluation of the scientific content of the paper, I would like to know the opinion of the authors about the need (or not) of reverting the "stuck-in-the-middle" authorship tendency in collaborative health research between high and medium-low income countries, and if yes, how this might be done.

Reviewer #4 (Remarks to the Author):

GENERAL COMMENTS:

In this well-written manuscript, the authors use robust epidemiological and parasite genomic data to compare *Plasmodium falciparum* (Pf) and *Plasmodium vivax* (Pv) infections in Guyana. They perform whole-genome sequencing on a large sample set collected from across the country and over the course of several years. The size and scope of the study is an important strength, providing the most in-depth analysis of the genomic epidemiology of these species in South America to-date, to my knowledge. Their results are consistent with expectation – that Pv parasites had higher genetic diversity but less relatedness in comparison to Pf parasites. While these findings are not a surprise, the scale of the analysis makes the work noteworthy. The authors include two paragraphs in the Discussion that attempt to translate their findings into concrete opportunities for the malaria program. This is important and much appreciated.

MAJOR COMMENTS:

1. Their finding that the majority of Pf and Pv infections involved non-mobile, local “infection sources” based on epidemiological data (Supp Fig 12) is highly relevant to malaria control efforts. Figure 5 attempts to bring together the genomic and epidemiological data, showing that the proportion of cases linking two geographical regions correlates with the proportion of parasites with IBD >0.5. The authors include several concrete examples of how their findings could be used by the malaria control program (Discussion - lines 391-421). Do the authors think a “universal radical cure” strategy like that recently reported by Thriemer et al. (PMID: 37979594) could have value in Guyana? Has Pv infection after Pf treatment been studied in Guyana? More broadly, are there other opportunities for “unified” (Pf and Pv) strategies here?

2. Key details relevant to sequencing and analysis need to be added to the Methods section. The authors achieved good sequencing results from DBS samples, generating analyzable genomes from the majority. The parameters used to determine whether a genome had sufficient coverage for further analysis (e.g. 80% at $\geq 3x$) need to be clearly defined. In addition, more details about how DNA from DBS was enriched for Pf and Pv need to be provided. For Pf, was the method described by Oyola et al. followed exactly or with modification? For Pv, the reference provided does not describe any oligos or relevant methodology. This needs to be rectified. For both Pf and Pv, did failure to generate an analyzable genome associate with low parasite density? If so, was selection bias introduced into your analyses? For bioinformatic analysis and figures, all code should be made publicly available through github or similar (with link provided in the manuscript).

3. Analysis of transmission focuses on epidemiological and spatial covariates. What about drug-resistance markers? Are any of the clonal Pf groups associated with drug-resistance alleles?

MINOR COMMENTS:

1. Results: How much sequencing effort was applied and what Pv/Pf coverage was achieved for individual samples (BioSample IDs)? Could a supplementary table be added with this information? This will make the data more accessible to other researchers.

2. Results: In the COI analysis, how were outliers processed? Was there any correlation with sequencing depth? THE REAL McCOIL can generate falsely elevated COI values in the setting of very low or very high coverage.

3. Results: The Pv clone recurrence findings are interesting, especially given that they were identified at HSPH in a blinded fashion and confirmed by the program in Guyana.

4. Line 123: “Until 2019” implies that *P. vivax* was overrepresented until this date, when something changed. I think the authors mean to say “throughout the study period”.

5. Fig 1: Before showing a map of Guyana, it would be nice to have a zoomed out version of its location within South America so readers who are not familiar with South American geography can understand its borders and overall location.

6. Fig 1B: Why are people with Afroguyanese ethnicity more likely to have Pf than other groups? Because they are more likely to live in areas where Pf is more common? Are there other possible explanations?

7. Fig 1C caption (line 565): did the authors mean "x-axis are ordered by P. vivax count.." not y-axis?

8. Fig 2: Could major mining regions be highlighted on the map? The authors discuss the role of mining on malaria transmission. Curious to see if this is reflected in the spatiotemporal data displayed here.

9. Lines 443-446: The concluding sentence implies that the disproportionate investment in Pf needs to be rectified. Given that the vast majority of global deaths are due to falciparum malaria, one could make a strong argument for continued "disproportionate" investment in Pf.

10. The authors should provide a competing interests statement (it was not visible to me).

Reviewer #5 (Remarks to the Author):

Dear Reviewers,

We would like to thank you for your very meticulous and constructive feedback. Your review has been very helpful in enhancing the manuscript. All points are addressed below in **green highlighting**, with line numbers referring to the change-tracked version of the manuscript.

Thank you again for your time,

Philipp Schwabl and co-authors

REVIEWER COMMENTS

Reviewer #1 (Remarks to the Author):

NCOMMS-24-21960

As many regions make significant reductions in malaria incidence globally, efforts to reduce the incidence of *P. vivax* have generally proven less successful than those for *P. falciparum*. Various *P. vivax* biological features like relapsing hypnozoites may be driving these challenges, but the epidemiological effects of these differences have been understudied due to a dearth of analyses in spatiotemporally matched genomic samples from sympatric populations. This manuscript contrasted the patterns of genomic relatedness and spatial structure between sympatric populations of two species in Guyana to assess the potential need for species-specific interventions. *P. vivax* isolates were more diverse, exhibited fewer clonal groups, and had shorter isolation-by-distance signals than *P. falciparum*. The frequency of highly related *P. vivax* infections correlated less closely with epidemiologic travel events than with *P. falciparum*. *P. vivax* had a higher polyclonality rate, which is a likely mechanistic contributor to the differences in diversity and relatedness in this species compared to *P. falciparum*. This study makes a meaningful contribution to the field, presenting an analysis of novel sympatric *Plasmodium* populations, providing important insights into the differences in population and spatial genetic structure, and offering potential implications of these insights for malaria elimination efforts in the country. Study limitations, some moderate and some minor, are noted below.

Lines 178-179: The manuscript notes that data from 250 polyclonal infections were excluded from the analysis; however, it does not provide the number of samples excluded for each species or final sample sizes for both species after removal of polyclonal infections. This information can be calculated using the percent polyclonal isolates noted later in the text or the supplement, but it would be good to have a clearer final sample size for each species reported earlier in the main text.

We have amended line 187 to specify that 598 *P. falciparum* and 548 *P. vivax* samples were identified as monoclonal and used for identity-by-descent analysis.

In addition, since polyclonal infections are more prevalent among *P. vivax* infections, it would be good to discuss how exclusion of more *P. vivax* isolates from the analysis impacted downstream inferences. Also, why not try to salvage some of the polyclonal samples that have a predominant clone?

We did seek to salvage predominant clones from biconal *P. vivax* samples (n=144) based on read-depth ratios, but we generally either observed too much within-sample variation in minor allele frequency (MAF) values or allele ratio modes too close to 0.5 (i.e., balanced strain ratios). For example, in the random set of 15 biconal *P. vivax* samples plotted below (Fig. R1), only G4G152, G4G196, and G4G432 show a low frequency mode around 0.15 – potentially amenable to phasing but still not confidently so. Most other samples show no clear frequency mode or a mode too close to 0.5 (e.g., G4G416 and G4G465). Frequent non-modal within-sample MAF distributions may partially reflect stochasticity of sWGA amplification and relatively modest *P. vivax* read-depths we have obtained.

Given little potential to phase based on read-ratios, the other possible approach would be to attempt population-based phasing, i.e., computationally separating out haplotypes based on linkage patterns (IBD segments) observed in monoclonal samples. Given the highly different linkage decay patterns (Supp. Fig. 20) and median relatedness between *P. falciparum* and *P. vivax* in Guyana, however, this approach would be vulnerable to species bias.

Fig. R1. Read-depth and within-sample minor allele frequency distributions for a random subset of 15 *P. vivax* samples classified as biconal using THEREALMcCOIL. Analysis uses read-depths of bi-allelic heterozygous SNP calls at sites in which the within-population minor allele frequency exceeds 0.02.

We do not think that the exclusion of more *P. vivax* than *P. falciparum* samples significantly impact downstream inferences as we are not aware of any evidence from the literature that specific parasite genetic variation is associated with propensity to occur in monoclonal vs. polyclonal infections. It could be argued that polyclonality exclusion would create smaller sampling representation for high-transmission areas, but a significant relationship between spatial variation in malaria incidence and polyclonality was not found in this study. High *P. vivax* dispersibility likely undermines location-specific parasite diversity distribution in the mining-associated transmission context of Guyana.

Please note that sampling exclusion also occurred on the basis of sequencing success, which we show moderately correlates with *P. vivax* parasitemia (qPCR) in response to Reviewer #4. We explain within that response why we also do not consider parasitemia-correlated sample exclusion as a key confounder of analysis. The overriding bias is the use of a passive sampling system for two parasite species which are associated with different parasitemias / detectability via microscopy/RDT, pyrogenic thresholds, host immunity and morbidity, etc. – this fundamental complexity in describing malaria cannot be well mitigated or accounted for by most any study design. The use of the passive approach and possible bias against low-parasitemia infections is indicated the limitations section (lines 468-479).

It is not clear why the authors chose to dichotomize IBD levels into >0.5 IBD sharing or not. Figure 3a indicates a wide range of total cumulative IBD sharing values, particularly for *P. falciparum*, that could be used to treat IBD as a quantitative variable, or at least a more granular ordinal variable, rather than dichotomous. A lot of information is lost by treating the data as dichotomous, which could provide better resolution, particularly for *P. vivax*, which had very few infections sharing >0.5 IBD. The authors note in the discussion that examining only outlier IBD relationships is a limitation, but do not provide a justification for why the analysis was performed this way.

We chose the 0.50 IBD cut-off because 1) it is epidemiologically meaningful (it identifies siblings and clones which may carry important epistatic alleles) and 2) least ambiguously relates to recent ancestry (population connectivity) considering the relatively inbred malaria transmission context of Guyana (IBD mean = ca. 0.311 for *P. falciparum*). Perhaps most importantly, the 0.50 IBD cut-off is equally well interpretable for both analyzed species, which differ strongly in the density of IBD values around the median (i.e. variance), presumably due to differences in standing genetic diversity as well as outcrossing rates.

Lower IBD cut-offs such as 0.375 or 0.40 do generally lead to similar inference, e.g., in the case of isolation-by-distance (Fig. R2 - now included in the manuscript as Supplementary Fig. 13), and may at times provide better resolution of gene flow (e.g., see Fig. R3 and its legend), but quantitative summarization of the distribution below these cut-offs incorporates a range in which IBD signatures of recent ancestry and more ancient population history overlap (see standard deviation (grey dashed lines) in Fig R2b). This complicates interpretation and generates larger residuals in relationships detected with respect to spatiotemporal variables.

Fig. R2. Relatedness vs. spatial distance using alternative dichotomization cut-offs (a) and using absolute IBD (b). Solid colored bars represent 90% confidence intervals generated by bootstrapping 1000x. In plot b, grey dashed bars represent +- standard deviation from mean IBD (points). Extended legend is provided in the newly added Supplementary Fig 13.

Fig. R3. Example of full-IBD distribution for pairwise sample comparison within and between selected epidemiological zones. Population differentiation most clearly manifests as outlier IBD frequency in both species (e.g., >0.50 IBD). In *P. falciparum*, additional valuable relatedness information does occur throughout the IBD distribution, but this is difficult to evaluate for *P. vivax*.

The patterns of isolation-by-distance shown for *P. falciparum* shown in Figure 3b are very strange with almost a cyclical pattern in specific distance bins. The authors suggest that this pattern may be driven by clonal outbreaks in commonly visited locations within these distance ranges. It would be helpful for the authors to discuss this observation in more depth. Do the epidemiological data support their hypothesis?

We suggested that prominent “upticks may reflect accumulation of relatedness between common travel hubs or mining foci and discontinuous transmission risk during interregional host movement.” We have amended this sentence slightly (lines 238-241) and added a ‘Supplementary Text 2’ section + Supplementary Fig. 14 following additional assessment which, yes, does provide supporting evidence, but also illuminates other complexities.

Specifically, we focused on a closer analysis of the prominent uptick in >0.50 IBD frequency observed at 100 - 130 km in Fig. 3b (x-axis position = 100 km). We first decreased window size to 9 km (previously 30 km) and step size to 1 km (previously 10 km) in order to pinpoint the uptick range more specifically as 111 - 132 km. Next, we plotted all site comparisons representing 111 - 132 km for which values of >0.50 IBD were observed in the genetic dataset (Fig. R4 map – now also included in the manuscript as Supplementary Fig. 14).

Fig. R4. Geographic comparisons underlying the *P. falciparum* isolation-by-distance slope irregularity at 111 - 132 km and consistent patient travel signals in the epidemiological database. In the map, segments represent infection locality pairs for which non-zero >0.50 IBD frequency occurs. Segment widths indicate the fraction of comparisons exhibiting >0.50 IBD and opacity levels indicate the absolute number of comparisons exhibiting >0.50 IBD. Localities are colored according to epidemiological zones. Relatedness elevation appears pronounced between the Lower Potaro and Lower Cuyuni / Lower Essequibo zones as well as between the Upper Cuyuni and Lower Cuyuni zones (underlined labels). Histograms at right indicate that non-zero distances between infection and diagnosis localities recorded in the 2019 epidemiological database also often fall within the 111 - 132 km range when cases represent prior stay (i.e., inferred infection) at prominent nodes of genetic connectivity in the map. Apart from excluding non-mobile cases (i.e., cases for which infection and diagnosis localities match), histograms also exclude cases diagnosed in West of Georgetown, Greater Georgetown, and East of Georgetown zones (considered non-endemic). The 111 - 132 km distance signal in the epidemiological database is generated primarily by cases diagnosed in Bartica Village District (98% for the Lower Potaro zone, 94% for the Upper Cuyuni zone, and 100% for Kuribrong and Waiamu River localities).

Index	Segment	Comparisons >0.50 IBD	Total comparisons	Fraction >0.50 IBD
1	Baramalli_River_vs_Cuyuni_-_Mazaruni_Reg_or_Reg_No_7	1	2	0.5
2	Big_Hope_River_vs_Cuyuni_-_Mazaruni_Reg_or_Reg_No_7	1	2	0.5
3	Potaro_River_vs_Rock_Creek	1	2	0.5
4	Bartica_Village_District_vs_Kuribrong_River	2	4	0.5
5	Matthews_Ridge_vs_Toroparu_River	4	8	0.5
6	Arimu_Mine_vs_Kuribrong_River	8	20	0.4
7	Big_Hope_River_vs_Serenamu_River	1	3	0.33333333
8	Arimu_Mine_vs_Potaro_River	3	10	0.3
9	Kuribrong_River_vs_OKO_RIVER	10	38	0.26315789
10	Cuyuni_-_Mazaruni_Reg_or_Reg_No_7_vs_Issano_Junction	1	4	0.25
11	Cuyuni_-_Mazaruni_Reg_or_Reg_No_7_vs_Rock_Creek	1	4	0.25
12	Cuyuni_-_Mazaruni_Reg_or_Reg_No_7_vs_Saint_Elizabeth_Creek	1	4	0.25
13	Aurora_7_vs_Karau_River	3	12	0.25
14	Arimu_Fall_vs_Cuyuni_-_Mazaruni_Reg_or_Reg_No_7	2	10	0.2
15	Kuribrong_River_vs_Mazaruni_River	2	10	0.2
16	Arimu_Mine_vs_Serenamu_River	6	30	0.2
17	Issano_vs_Isseneru	1	6	0.16666667
18	Matthews_Ridge_vs_Tamakay	1	6	0.16666667
19	Aurora_7_vs_Issano	2	12	0.16666667

Table R1. Values underlying segments mapped in Fig. R4.

In the Fig. R4 segment mapping (also see underlying >0.50 IBD frequency in Table R1), we principally observe:

- Several axes fanning out from the Lower Essequibo zone, especially:
 - Between Bartica and the Lower Potaro zone (e.g., Kuribrong River locality)
 - Between Bartica and the Upper Cuyuni zone (e.g., Waiamu River locality)
- Several axes fanning out from the Lower Cuyuni zone (e.g., Arimu, Oko River locality)
 - especially North-South connection to the Lower Potaro zone
- Several axes fanning out from Puruni River locality

As a sanity check, we can confirm the strong uptick contribution of Lower Essequibo zone, Lower Cuyuni zone, and Puruni River (central nodes of segment fanning) by re-running sliding window isolation-by-distance analyses excluding any comparison involving these terms. This smooths out most prior irregularity in slope (Fig. R5).

Fig. R5. Isolation-by-distance plot for *P. falciparum* sample excluding comparisons involving Puruni River locality or sites in the Lower Essequibo or Lower Cuyuni zones (window size = 30 km and step size = 10 km (as originally applied in Fig. 3b)).

The observed genetic connectivity elevations between the relatively distant (111 - 132 km) site pairs may mean that co-visitation of these distant site pairs is elevated, increasing pairwise relatedness over time, especially if clonal blooms occur and clones / clone descendants remain successful/maintained at both nodes. Also, the means of travel by which co-visitation occurs may not always allow for continuous transmission opportunities in the intervening landscape (e.g., mosquitos not active on windy boat / on the broad Essequibo or during road travel between the Lower Cuyuni and Lower Potaro zones), enhancing the relatively discrete uptick/interruption to isolation-by-distance slope.

To assess this hypothesis, we can screen the 2019 epidemiological database as to whether patients' travel history frequently implies 111-132 km travel for cases with inferred infection sites representing the most prominent nodes mapped in Fig. R4a / tabulated in Table R1 (especially the Lower Potaro zone (localities such as Kuribrong River) and the Upper Cuyuni zone (localities such as Waiamu River)). If we plot the distribution of distances to sites of diagnosis for mobile infections attributed to these prominent nodes, we find that distances frequently fall within the 111 - 132 km range, and much of this signal represents movement to diagnosis in Bartica (Fig. R4b).

Therefore, the epidemiology data from 2019 do direct for elevated co-visitation of zones such as Lower Potaro + Lower Essequibo and Upper Cuyuni + Lower Essequibo, and indirect evidence for co-visitation of zones such as Upper Cuyuni + Lower Cuyuni (because Lower Cuyuni is directly adjacent to Bartica).

The epidemiological database does not, however, mirror >0.50 IBD segment patterns representing the North - South Lower Potaro - Lower Essequibo zone comparisons or those involving Puruni River locality. In fact, the database contains zero cases of host movement from Lower Potaro zone to Lower Cuyuni zone (or vice versa) and only 4 total cases involve 111 - 132 km movement to or from Puruni River. Incongruences between parasite IBD-based connectivity signals and the patient travel history information may reflect parasite relatedness patterns unrelated to contemporary gene flow (e.g., historic, geographically heterogeneous clonal expansions). The limited resolution at which patient movement could be characterized using the

epidemiological database likely also plays a role. Given just two recorded localities per patient, details on transit routes and stop-over sites intervening infection and diagnosis events are hidden from view. It is very possible, for example, that many cases representing infection in LP and diagnosis in Bartica involve patients which did not initiate travel to Bartica or associated areas for the purpose of medical attention. Mahdia, the capital of Region VIII, is a much easier option within LP when medical attention is the goal. Bartica is relatively easily accessible from LP (e.g., <4 hours by land according to the Ministry of Infrastructure), and represents the primary logistics and departure hub for mining activities along the Mazaruni and Cuyuni Rivers. Examples of relatively accessible mining areas include Oko River and Arimu Mine (see nodes in Fig. R4a). An infected host from LP working in such areas accessed in Bartica is very likely to return to Bartica for medical attention once symptoms arise.

Did they conduct a sensitivity analysis, using different sized distance bins, or excluding identical clones, to try to better understand this pattern? Also, were genomes sampled from the same person at different time points excluded from the relatedness analyses, since these are not independent infections?

The original analysis for *P. falciparum* used window size 30 km and excluded comparisons involving clones represented by ten or more individuals (as indicated in figure legend). The exclusion represented an attempt to dampen potential connectivity signals that actually represent long-term and geographically widespread establishment of large clonal groups as opposed to more recent dispersal events (active gene flow).

We repeated analysis using larger window sizes (50, and 75 km), removing the filter against large clonal groups, and changing the step size of the window slide (Fig. R6). As would be expected, increasing window size begins to smoothen irregularities in the isolation-by-distance relationship, but the general trend remains robust. Removing the filter against large clonal groups slightly enhances the irregularities discussed previously (e.g., uptick around $x = 110$ km) and increases error width at the largest distance classes. These observations are consistent with our previous inference that relatedness accumulation among selected localities create connectivity signals that interrupt the global isolation-by-distance relationship. These upticks are relatively discrete (e.g., see response above relating specifically to the 111 - 132 km distance class) but appear more rounded due to sliding window overlap (e.g., compare Fig. R6a to Fig. R6e).

Fig. R6. Isolation-by-distance analysis with alternate window size and step size parameters as well as the inclusion of super-clone member samples.

We should further note that relatedness accumulation / genetic connectivity deviations among selected localities are not primarily being detected in the form of clonal relationships (i.e., >0.90 IBD) and due to the inclusion of multiple clone representatives per group (we allowed all member samples for clonal groups of size 2-9 in the original analysis) as Fig. R6d might be interpreted. We rejected this interpretation based on three further isolation-by-distance analysis changes (Fig. R7):

- a) omitting comparisons involving clones allows only one clonal representative per clonal group
- b) omitting clones representing clonal groups with 3 or more member samples
- c) quantifying >0.40 to <0.80 IBD frequency instead of >0.50 IBD frequency.

These changes should remove signals stemming primarily from the detection of >0.90 IBD clones. They do not significantly change previously observed irregularities in isolation-by-distance slope.

Fig. R7. Isolation-by-distance plot for *P. falciparum* samples a) using only one representative clone per clonal group, b) excluding clones from clonal groups represented by 3 or more members, and c) classifying elevated relatedness (y-axis) as the fraction of comparisons representing >0.40 IBD and <0.80 IBD.

The original analysis for *P. vivax* used window size 30 km and did not exclude comparisons involving the 11 patients identified as relapsing individuals in our posthoc analysis. We did not exclude these patients because relapse represents an important type of gene flow in *P. vivax* which we should not obscure. Nevertheless, excluding comparisons involving these patients does not significantly change the relationship between spatial distance and the frequency of pairwise >0.50 IBD (Fig. R8).

Fig. R8. Isolation-by-distance plots for *P. vivax*, with/without excluding 11 visits identified as relapses.

It would be helpful to know the average read depth of the sequence data from *P. falciparum* and *P. vivax* and how that might have impacted inferences of COI. I anticipate that parasitemias (and thereby read depth) might be greater for *falciparum*, which could make the reported COI differences between the species less pronounced (may more easily identify minor clones in *falciparum* infections than in *vivax* infections).

This is indeed true. The mean read-depths for mapped *P. vivax* sequence reads (information now added to Supp. Table. 2) are statistically lower than those for *P. falciparum* (Welch's t-test, $p < 0.001$). They also differ significantly for monoclonal vs. polyclonal samples in *P. vivax* (Welch's t-test, $t = 2.777$, $p = 0.006$), but not in *P. falciparum*. Relationships between COI values (1, 2, 3, 4) and read-depth are not significant within either species analysis, although in *P. vivax* analysis the comparison COI = 1 vs. COI = 3 approaches the significance threshold 0.05 (Tukey's HSD test, $p\text{-value} = 0.058$).

We have therefore added Supplementary Fig. 19 (= Fig. R9 below) and main text lines 325-326 that the species differences in COI indicated by our study may be conservative due to lower sensitivity to detect minor clones in *P. vivax*. Underdetection may apply primarily to >2 COI calls based above Tukey HSD test.

Fig. R9 Relationship between complexity of infection (COI) values and sequencing depth in *P. falciparum* and *P. vivax* samples. Boxplots summarize variation (median and quartiles) in mean read-depth associated with each sample (points). Samples are grouped by COI level (1 - 4) along the x-axis. One outlier *P. falciparum* sample (G7B667, average read-depth = 262.9) is not shown. Mean read-depth differs significantly between species (Welch's t-test, $t = 20.324$, $p < 0.001$) and for monoclonal vs. polyclonal samples in *P. vivax* (Welch's t-test, $t = 2.777$, $p = 0.006$), but not in *P. falciparum*. Mean read-depths do not differ significantly by COI level within either species (Tukey's HSD test).

It would also be helpful for the authors to discuss in the methods how the infection location was designated, i.e., was it designated based on location of diagnosis, or for individuals who traveled, based on travel location?

The Methods section lines 503-512 have been amended to: Infection localities were designated by asking 'Where [the] patient stayed 2 weeks ago' (see survey form in Supplementary Fig. 23). Each voluntary travel history response was classified to one of 460 country-wide localities with known latitude/longitude coordinates. Infection localities were entered as missing (NA) if patients provided no response on prior stay (i.e., infection localities were not equated to diagnosis localities unless patients specified having stayed in the same locality 2 weeks ago). For foreign localities of prior stay (e.g., in neighboring Venezuela), geographic specifics were not retained in the database. The prior stay field simply shows the term 'imported' for these cases, which we re-code as 'foreign' in our analysis).

Figure 5 – The correlation between IBD >0.5 and case flow is higher for *P. vivax* than for *P. falciparum*, but this may be strongly driven by one outlier observation (#6), because overall, the correlation appears to be lower for *P. vivax* than *P. falciparum*.

We now note in the main test that the *P. vivax* regression is outlier-dependent (line 307). We added to the figure legend that “significant correlation is lost from the *P. vivax* regression when outlier point #6 is omitted (Pearson’s $r = 0.07$, $p = 0.675$). Significance remains when omitting this point from the *P. falciparum* regression (Pearson’s $r = 0.41$, $p = 0.045$).”

Other minor points:

Figure 1. Figure legend does not explain the final map on the left of the top row that defines the epidemiological zones. In the text (line 138), figure 1C is referred to and seems to reference the map with epidemiological zones, but 1C is currently the graph of cumulative cases. The numbers of the epidemiological zones are never directly linked with the names used in the x-axis of Figure 1C, making it confusing to determine where on the map the authors are referring to in later discussions of results.

We have changed line 141 to indicate “Fig. 1 map” instead of Fig. 1c. The zone numbers used in the map occur as point labels in Fig. 1c. We have increased the font size as much as possible for clarity.

Figure 1E: What is the quantitative estimate of significance?

Dunn (1964) Kruskal-Wallis multiple comparison; p-values adjusted with the Benjamini-Hochberg method.

Comparison in Females	P.adj
<15 vs. 15 - 55	0.003891597
<15 vs. 55+	0.121101190
15 - 55 vs. 55+	0.143441509

Comparison in Males	P.adj
<15 - 15 vs. 55	3.525073e-03
<15 vs. 55+	4.637637e-06
15 - 55 vs. 55+	7.786632e-02

We have amended the figure legend: “Asterisks indicate significant differences for age bins <15 vs. 15-54 (bf-adj. $p = 0.004$) and <15 vs. >55 (bh-adj. $p < 0.001$, Dunn test).”

In Supp. Fig 12, it is not clear from the legend how “Foreign Import” cases were defined and how that differed from non-local cases.

We have clarified in the Methods section (lines 501-512):

“Infection localities were designated by asking ‘Where [the] patient stayed 2 weeks ago’ (see survey form in Supplementary Fig. 23). Each voluntary travel history response was classified to one of 460 country-wide localities with known latitude/longitude coordinates. Infection localities were entered as missing (NA) if patients provided no response on prior stay (i.e., infection localities were not equated to diagnosis localities unless patients specified having stayed in the same locality 2 weeks ago). **For foreign localities of prior stay (e.g., in neighboring Venezuela), geographic specifics were not retained in the database. The prior stay field simply shows the term ‘imported’ for these cases, which we re-code as ‘foreign’ in our analysis).**”

There are no epidemiological region numbers indicated on Fig 4 or Supp Fig 12, which makes some of the patterns described in the main text difficult to discern, e.g., reduced connectivity between Region I and Regions VII and VIII.

We have added labeling and outlines for Regions I/VII/VIII/IX and for bordering countries to Fig. 4 and what is now Supp. Fig. 16.

Is there a justification for the use of a 2% MAF cutoff? Many studies use 1% as a MAF cutoff (which is arbitrary), but I wasn't sure if there was a specific rationale here for using a more stringent cutoff, when having more diversity may provide better resolution.

We used MAF threshold $\geq 2\%$ to account for the occurrence of less common allelic variance whilst minimizing erroneous calls, especially given lower mean depth in *P. vivax*. We also considered 2% appropriate for the geographic scale of our study, which also involved a Venezuelan sample subset. A lower threshold ($\geq 1\%$) used by Vanhove et al. (PMID: 38352461) for *P. falciparum* WGS in Guyana resulted in a very similar IBD distribution (mean = 0.283 (0.111 sd) vs. mean = 0.311 (0.125 sd) in this study), probably that study's result being slightly lower due to IBD calculation across a 5-year sample set (2016-2021) as well as a suspected typo in that publication (median reported as mean). Another pop. gen. study by Carrasquilla et al. involving *P. falciparum* in Guyana (PMID: 36542676) used a $\geq 5\%$ threshold for IBD estimation.

In Supplementary Figure 8, semi-transparent symbols are hard to see, and group IDs are difficult to distinguish by color (e.g., 19, 31, 23, 33).

We have amended to distinct point colors (primarily from <https://sashamaps.net/docs/resources/20-colors/>) and reduced point sizes in what is now Supp. Fig. 10. We changed the semi-transparent symbols to full transparency given that we did not aim to demonstrate all 79 clonal groups in this figure. The point jitter seed now also matches that of Supp. Fig. 11.

Supplementary Tables 1 and 2 are flipped from the notations in the text. Table 2 currently corresponds to the 2019 infections, and Table 1 corresponds to the metadata for sequenced isolates.

Thank you! We have amended Supplementary Table names (1>2 and 2>1).

Reviewer #2 (Remarks to the Author):

Reviewer #3 (Remarks to the Author):

The manuscript by Schwabl et al entitled "Contrasting genomic epidemiology between sympatric Plasmodium falciparum and Plasmodium vivax populations" uses standard epidemiological data and genomic features to compare the epidemiology of *P. vivax* and *P. falciparum* in Guyana. The study gives a comprehensive and complete comparison of Pf and Pv population genomics, highlighting the main differences found and connecting them with suggestions for control and elimination strategies. The results show how genomic analysis can identify epidemiological features that are not observable from traditional symptomatic case

analysis. The results are relevant, robust and well presented. The authors present many possible interpretations for the results, although these are hard to prove.

Major comments:

There is a chronological discrepancy between the standard epi data (mainly 2019) and genomic data (2020 and 2021). Why did authors decide to focus on epi data from 2019 and not from 2020/2021? Figure 1a shows a decline of malaria cases starting in 2019 that may affect the results (ie, relating polyclonality in samples from 2020-21 with incidence data from 2019; Fig 6b).

The most recent curated epidemiological dataset at the case level represents 2019, which unfortunately leads to a discrepancy between the timeframe the genomic samples were obtained and the year for which routine epidemiological records are available. We do not think this generally prevents investigating the epidemiological and genomic data together in this paper, as most of the epidemiological features which we have focusing on (e.g., principle host movement trajectories along rivers/roads, host age/sex/ethnicity/nationality composition, intra-annual malaria seasonality (Supp. Fig. 1), mining-associated malaria foci (tied geographically to mineral resources (new Supp. Fig. 5))) are expected to remain relatively stable for 2019 vs. 2020/21. As the reviewer importantly highlights, the comparison of estimated incidence to polyclonality rate variation among epidemiological zones may represent an exception where both data types should be as tightly possibly temporally matched.

We have therefore inquired with the Guyana Ministry of Health for the release of additional 2020-21 case count information. The Ministry assembled a list of travel history-annotated aggregate case counts matching the genomic sampling period. These are case totals recorded without separation according to species and no case-level details like those available for 2019 but nevertheless can address the matter of possible inconsistency in geographic transmission intensity variation between years.

We used this data to recreate the plot comparing estimated incidence to polyclonality rate variation among epidemiological zones (with an additional amendment to the previous method which unnecessarily divided the incidence estimate by raster cell count). This plot has now become Supp. Fig. 21. The plotted trends are very similar when using the 2019 vs. 2020-21 epidemiological datasets because geographic case count variation is highly correlated between these years (Fig. R10 below).

In any case we have decided to move 6b from the main manuscript to the supplement (Supp. Fig. 21) because the non-significant trends add little extra value (no clear inference) and because our estimation incidence uses a new and not easily validated approach (normalization of infections attributed to each zone by a population density estimate obtained from the LandScan Project).

Fig. R10 Both axes represent total counts (i.e., not specific to malaria species) because species-specific counts are not available for 2020-21. Red: epidemiological zones originally used in Fig. 6b based on our genomic sample size threshold of 20.

Regarding the figure originally numbered as 6c (polyclonality rate vs. case fraction by gender) which now becomes 6b, this uses gender data from the genetic database (i.e., no temporal discrepancy between x and y axes). This is now clarified in the legend. In any case, as also stated in the legend, gender fractions are highly similar in the 2019 epidemiological database (e.g., for the three NDCs analyzed in the plot, *P. falciparum* male case fraction ranges 0.732-0.748 in 2019 vs. 0.687 to 0.765 in 2020-21 and *P. vivax* male case fraction ranges from 0.649-0.680 in 2019 vs. 0.693 to 0.722 in 2020-21).

Most of the analysis is centered on relatedness and polyclonality, which are highly sensitive to how informative of diversity are the used markers (SNPs) for each species. The results of this study point to higher diversity, polyclonality and lower relatedness (IBD) in Pv than in Pf: could this be a consequence of the chosen SNPs (ie, especially diverse for Pv compared to Pf, not necessarily meaning that Pf strains have a higher "real" IBD)? In particular, the analysis is done for 337 SNPs for Pf versus 350 SNPs for Pv: could this imply lower IBD and higher polyclonality statistics from Pv than from Pf just by statistical chance? Also, the filtering of >10% minor allele frequencies could imply different exclusions for the different species, especially if one has a larger presence of low frequency alleles. Can authors think on a way to to prove that the comparisons of the study are not biased due to the selection of markers (SNPs)?

We verified COI calling robustness in response to this important query.

Within each species, we calculated read-depth mean and standard deviation among samples for each of the random SNP sites applied in the original COI calling analysis (THEREALMcCOIL) and then filtered out sites with mean ≥ 200 or sd ≥ 150 to address the possibility that high depth and/or variability confounds estimation. This filtering also conveniently reduced the SNP sets to equal size ($n = 327$) for both species.

We then re-ran COI analyses using the original ($n = 337$ and $n = 350$) SNP sets and the new filtered ($n = 327$ and $n = 327$) SNP sets. The original SNP sets produced identical calls to the original analysis (sanity check) and the new filtered SNP sets produced nearly identical calls to the original analysis: 674/675 (99.9%) matches for *P. falciparum* and 711/721 (98.6%) matches for *P. vivax*. Mismatched calls in *P. vivax* were as shown below (3 originally polyclonal now monoclonal, 6 originally monoclonal now polyclonal, and 1 consistently polyclonal but originally with 1 less strain).

The $\geq 10\%$ minor allele frequency (MAF) threshold involved in the COI calling SNP set was chosen given the

objective to address species differences in the likelihood for strains to co-occur within infections (which is fairly described by common variants within each species) as opposed to sensitivity towards population diversity differences (e.g., frequency of rare variants) between species.

For these reasons we have decided to keeping the COI methods as originally applied. We also note having assessed call consistency in lines 589-591.

P. vivax sample	New COI call using 327 filtered SNPs	Original COI call using 350 filtered SNPs	Difference between calls
CEM566_Pv-5	1	2	+1 higher in original analysis
G1P004	3	2	-1 lower in original analysis
G4G1434	1	2	+1 higher in original analysis
G4G1690	2	1	-1 lower in original analysis
G4G1831	2	1	-1 lower in original analysis
G4G951	2	1	-1 lower in original analysis
G4G006	2	1	-1 lower in original analysis
G4G378	1	2	+1 higher in original analysis
G4G398	2	1	-1 lower in original analysis
G7B185	2	1	-1 lower in original analysis

Regarding IBD estimation, we used a lower MAF threshold ($\geq 2\%$) to account for the occurrence of rare allelic variance whilst minimizing erroneous calls, especially given lower mean depth in *P. vivax*. A lower threshold ($\geq 1\%$) used by Vanhove et al. (PMID: 38352461) for *P. falciparum* WGS in Guyana resulted in a very similar IBD distribution (mean = 0.283 (0.111 sd) vs. mean = 0.311 (0.125 sd) in this study), probably that study's result being slightly lower due to IBD calculation across a 5-year sample set (2016-2021).

Minor comments:

The results sections is accompanied by frequent interpretation of the data (ie, 151-154, 227-228, 266) which may be better allocated in the discussion section. Ç

We further shortened commentary within the Results section in lines 159-160, 258-268, and 351-352.

Authors explore the relationship between polyclonally and transmission intensity. Naturally immunity (and therefore intensity of transmission=exposure levels) shape parasite densities, which in turn can affect the ability to detect polyclonal infections (ie, high parasite densities, higher probability of detecting multiple genotypes if present). Do authors think this is something that should be taken into account in the analysis?

We did indeed consider that immunity is correlated with parasite densities and therefore may also affect sequencing success, read-depth, and the detection of polyclonality, especially COI levels greater than 2 (e.g., new Supplementary Fig. 19, see also response to Reviewer #1). We have not directly measured immunity in the study population, but do have metadata on certain demographic features – especially ethnicity, age, and sex – which we hypothesized may correlate to infection refraction/tolerance levels and manifest in differential read-depth or sequencing success. For example, Amerindians may have greater cumulative *P. vivax* exposure

(possibly leading to suppression of higher parasitemias and/or to less morbidity relative to other ethnicities experiencing the same parasitemias) due to immunity developed in home/childhood environments that may more often occur in malaria-endemic areas. Adults (15-54 years) may also have greater cumulative exposure than children (<15) or may have stronger immune health than older adults (>54). Third, the male sex may have different cumulative exposure than the female sex due to different occupational affinities or risk behavior. We tested for read-depth or sequencing/success failure to these factors using various statistical tests, both separately for each species and in a pooled fashion, but no significant differences are evident (also when omitting p-value adjustment for multiple comparisons). An example plot for read-depth vs. ethnicity group classification is provided below (Fig. R11).

Fig. R11 Mean read-depths for *P. falciparum* and *P. vivax* by ethnicity (non-significant variation among groups).

It is very possible that if relationships between immunity-associated demographics and sequencing yield / polyclonality detectability exist, we would not detect them via the passive, sentinel-site based sampling approach, which generally selects for higher parasitemias (symptomatic patients / infections above above pyrogenic thresholds) among which variation in sequencing success is less pronounced.

Lines 185-189: Why is >0.90 IBD the cutoff to define clonal groups? Why not 0.95 or 0.99, for instance? Is there a biological, statistical or referential reason for this?

Using the >90% IBD threshold, the majority (>97%) of detected sample pairs actually show IBD values >95% in either species. The biological interpretations do not change among these thresholds, but keeping the threshold at 90% allows for some differences due to genotyping error or *de novo* mutation, rates of which may differ between species.

Lines 122-123: The authors show the temporal trends of incidence from both species, showing fluctuations between their relative abundance, with some periods dominated by Pv and some other periods by Pf. It would be very interesting to know if the authors have any interpretation of hypothesis for these changes.

Generally the species rolling case count curves are very parallel/correlated over time, which we would interpret as additional evidence of high co-endemicity between species in Guyana, i.e. both species being subjected to the same environmental variability across seasons and years.

The curve inflection near 2013, with *P. vivax* becoming overrepresented in the last ca. 7 years for which data is available is consistent with other examples of regions nearing elimination goals (e.g., Solomon islands, Myanmar, Cambodia, Lao PDR) (lines 409-410) and the general concern that *P. vivax* is responding less well to intervention strategies that work against *P. falciparum* (lines 59-61 among others).

The magnitude of curve shift during year 2013 and the strong *P. falciparum* peaks in the final months of 2011 and 2012 are especially interesting. This *P. falciparum* peak has been associated with gold price peak in Salazar et al. 2021, Lancet Planet. Health (PMID: 34627477). Salazar et al. suggest monthly gold price as a primary driver of monthly *P. falciparum* cases in mining-related transmission regions based on a generalized linear modeling approach. They also identify a smaller but significant correlation between malaria abundance and the Southern Oscillation Index (SOI, 1-year shifted in the mining regions). A strong La Niña (positive SOI, generally associated with increased rainfall in South America) period occurred between March 2010 and April 2012 (<https://www.ncei.noaa.gov/access/monitoring/enso/soi>). We now add some of these details in lines 124-125.

We also briefly addressed relationships between malaria case numbers in Guyana and the major infectious disease prevalence changes in the bordering country of Venezuela (which generally started post-2012, PMID: 30799251). We plotted yearly reported malaria case numbers by species for Venezuela and Guyana using records from the Malaria World Report (Fig. R12). The data from Venezuela do not show the same 2011-2012 *P. falciparum* peak (now more of a bump due to log10 y-axis) nor the 2013 species majority flip seen in Guyana.

Fig. R12. Yearly reported *P. falciparum* and *P. vivax* cases based on the World Malaria Report (WHO). The bump in *P. falciparum* cases observed in 2011 and 2012 in Guyana is not paralleled in Venezuela. Large increases observed in Venezuela beginning ca. 2012 are also not paralleled in Guyana.

In general, but especially in Results: Species intricacies of a wide co-endemic distribution: are the p-values corrected for multiple comparisons?

In the main text we have highlighted statistics in pink which have been corrected for multiple comparisons. The methods used are now indicated (e.g., Bonferroni, Benjamini-Hochberg, Holm). We have highlighted statistics where multiple comparisons are not relevant in cyan. Finally we have highlighted two instances relating to COI in yellow where multiple comparisons do occur (lines 354 and 687-688), but the few comparisons tested were planned a priori and correction is arguable given small sample sizes. We now explicitly state absence of correction in these two cases.

Line 149-151: The authors mention an elevated case-bias towards non-Amerindian working-age males. Do we have a statistical significance for this statement?

Corresponding statistical tests and plots are noted in the previous three sentences. However, this statement may have imprecisely merged two separate findings (1) elevated adult-to-child case ratio irrespective of ethnicity and 2) less marked adult-to-child case ratio in Amerindians vs. in non-Amerindian ethnicities). We changed the statement to:

“Elevated case bias towards working-age males, more strongly so in non-Amerindian than in Amerindian ethnicities, is consistent with male-dominated hinterland mining activity driving malaria burden in Guyana^{30,32} (see mineral resource map in Supplementary Fig. 5), especially in non-resident (e.g., ‘coast lander’³³) mining groups” (lines 154-157).

Line 176: I would mention the name of the hidden state model here for clarity.

We now specify hmmlBD in line 186.

Line 178-179: I would mention more details of the Bayesian framework used to exclude the 250 samples.

We now specify THEREALMcCOIL in line 189 and provide further description in lines 576-591.

Lines 325-326: is the male representation coming from a possible bias in reporting of care seeking behaviour?

Our dataset does not allow for a direct analysis of potential reporting biases. Mining camps harbor most malaria cases – especially of *P. falciparum* – which we and others (e.g., Salazar et al. 2021, Lancet Planet. Health) think explains the higher proportion of cases in adult males. The underlying true case incidence in adult males might even be higher than recorded via the passive sampling approach, according to a study by Olapeju et al. 2020, PLoS One which suggests that malaria care-seeking rates in mining camps – which are mainly populated by male workers – remain low in Guyana. We have amended lines 124-125 and 154-161 near the beginning of the Results section to introduce the male-dominated nature of Guyana’s mining sector with three key citations (the two above + Hilson & Laing 2017, J. Rural Stud. on historical ethnic associations to itinerant mining work) as well as a Supp. Fig. 5 map illustrating the prevalence of mining deposits recorded for Western Guyana.

Lines 431-432: regarding limitations, the fact that IBD was conducted only on monoclonal infections could imply some bias in the statistics? This would imply a larger exclusion for Pv than for Pf, for example.

Final sample sizes remained similar to that for *P. falciparum* and we are not aware of an association between parasite diversity vs. propensity to occur within polyclonal infections linked to underrepresentation in analysis. It could be argued that polyclonal sample exclusion would reduce sampling representation for high-

transmission areas, but a significant relationship between spatial variation in malaria incidence and polyclonality was not found in either species in this study. We considered to use the Dcipher (PMID: 36000888) method, now cited in line 471, but the impact of differences in linkage decay (Supp. Fig. 20) and intra-host relatedness observed/expected between species is not yet fully established.

As authors mention, spatiotemporally matched sampling schemes are very rare and challenging to be done. Should this be better recognized in the distribution of authors? There seems to be a tendency toward middle positions from the malaria-endemic authors. While this is not an element that I consider as relevant for the evaluation of the scientific content of the paper, I would like to know the opinion of the authors about the need (or not) of reverting the “stuck-in-the-middle” authorship tendency in collaborative health research between high and medium-low income countries, and if yes, how this might be done.

We agree that the placement of LMIC authors in middle positions is unfortunate, given the essential contributions of endemic-country public health professionals and scientists to many publications, and thank the reviewer for raising this issue. We have attempted to address this issue in this manuscript by promoting one of our Guyanese co-authors (Collette Clementson) to co-first-authorship in recognition of her important role.

Reviewer #4 (Remarks to the Author):

GENERAL COMMENTS:

In this well-written manuscript, the authors use robust epidemiological and parasite genomic data to compare *Plasmodium falciparum* (Pf) and *Plasmodium vivax* (Pv) infections in Guyana. They perform whole-genome sequencing on a large sample set collected from across the country and over the course of several years. The size and scope of the study is an important strength, providing the most in-depth analysis of the genomic epidemiology of these species in South America to-date, to my knowledge. Their results are consistent with expectation – that Pv parasites had higher genetic diversity but less relatedness in comparison to Pf parasites. While these findings are not a surprise, the scale of the analysis makes the work noteworthy. The authors include two paragraphs in the Discussion that attempt to translate their findings into concrete opportunities for the malaria program. This is important and much appreciated.

MAJOR COMMENTS:

1. Their finding that the majority of Pf and Pv infections involved non-mobile, local “infection sources” based on epidemiological data (Supp Fig 12) is highly relevant to malaria control efforts. Figure 5 attempts to bring together the genomic and epidemiological data, showing that the proportion of cases linking two geographical regions correlates with the proportion of parasites with IBD >0.5. The authors include several concrete examples of how their findings could be used by the malaria control program (Discussion - lines 391-421). Do the authors think a “universal radical cure” strategy like that recently reported by Thriemer et al. (PMID: 37979594) could have value in Guyana? Has Pv infection after Pf treatment been studied in Guyana? More broadly, are there other opportunities for “unified” (Pf and Pv) strategies here?

This is a great suggestion. We do think universal radical cure has potential value in Guyana given levels of spatiotemporal co-endemicity seen in few other malaria regions of the world and the clear risk profile for co-infection (adults of non-African descent involved in the mining sector) described in this study (as now added to the Discussion in lines 454-460). The adult focus of the disease may also be relatively conducive to this strategy given that optimal pediatric dosage formulations for primaquine are less well established. We are however not aware of any prior study providing information on *P. vivax* infection after *P. falciparum* treatment in Guyana, and we could not comment on the frequency of this phenomenon ourselves due to blinding applied to the 2019 database.

2. Key details relevant to sequencing and analysis need to be added to the Methods section. The authors achieved good sequencing results from DBS samples, generating analyzable genomes from the majority. The parameters used to determine whether a genome had sufficient coverage for further analysis (e.g. 80% at $\geq 3\times$) need to be clearly defined. In addition, more details about how DNA from DBS was enriched for Pf and Pv need to be provided. For Pf, was the method described by Oyola et al. followed exactly or with modification? For Pv, the reference provided does not describe any oligos or relevant methodology. This needs to be rectified.

We added these details and provided clear primer set citations (Pf1-10 and Pvset1) in lines 546-558:

“We subsequently applied whole-genome amplification to DNA extracts, each 50 μ l sample reaction consisting of 5 μ l 10x phi29 polymerase buffer (NEB B0269S), 0.125 μ l (2.5 μ g) recombinant albumin (NEB B9200S), 0.5 μ l primer mix (10 oligos combined at 250 μ M, see below), 5 μ l 10 mM dNTPs (Thermo Scientific), 26.375 nuclease-free water, 3 μ l (30 units) phi29 DNA polymerase (M0269L), and 10 μ l DNA. Reactions were prepared on ice, with components added in the order shown. Primer mixes consisted of ‘Pf1-10’ for *P. falciparum*⁵⁴ and ‘Pvset1’ for *P. vivax*⁵⁵ (set selection based on previous microscopy or RDT-based species assignments). The 3’ regions of these primers contain phosphorothioate bonds. Amplifications were generated using step down incubation (35 °C for 5 min, 34 °C for 10 min, 33 °C for 15 min, 32 °C for 20 min, 31 °C for 30 min, 30 °C for 16 h, and 65 °C for 15 min), followed by cooling to 4 °C. We then applied AMPure XP magnetic beads (Beckman Coulter A63881) at room temperature to exchange post-reaction sample buffer to 10 mM Tris-HCl + 0.1 mM EDTA.”

Regarding sample exclusion rules, we added more detail in the joint call filtration text section:

“All downstream genetic analyses focused on SNP sites in core regions of the genome^{58,59} and >5 bp from any INDEL call. Samples missing genotype calls for >50% SNP sites at $\geq 2\%$ minor allele frequency (after application of the core region filter and INDEL masking steps) were also excluded from further analysis. Among excluded samples, coverage (% bases represented by >5 reads) averaged 12.9 in *P. falciparum* and 9.9 in *P. vivax*. Following sample exclusions, we removed sites at which >7.5% of monoclonal samples showed multi-allelic (‘heterozygous’) calls. These filtration steps generated the base SNP call sets for each species. Additional, analysis-specific settings are noted below.”

For both Pf and Pv, did failure to generate an analyzable genome associate with low parasite density? If so, was selection bias introduced into your analyses?

We do not have qPCR or microscopy-based parasite density metrics for the present sample set. However, we have qPCR-based *P. vivax* parasite density estimates (following Mangold et al 2009, J Clin Microbiol) for other field samples from Honduras which have been processed using the same extraction + sWGA + WGS protocol steps. We also have qPCR estimates for other sWGA’d field samples from Guyana, Colombia, and Venezuela genotyped using a multiplexed amplicon sequencing approach (‘Pv-GTseq’ – 249 targets). Obtaining 5+ reads for either >50% SNP sites (post-quality filtration and exclusion of non-coding regions) or for >50% Pv-GTseq targets generally appears to require >50-100 parasites/ μ l, but extensive failure/success variation still occurs above this threshold (Fig. R13). We do not consider the observed trends a major source of bias to this study as we are not aware of any evidence from the literature that infections featuring <100 parasites/ μ l vs. infections featuring >100 parasites/ μ l are associated with distinct parasite genetic diversity. Also, we should note that this study followed a passive sampling approach, i.e., only collecting samples from patients which independently sought medical care, likely due to malaria symptoms. Supposing a pyrogenic threshold of ca. 100 parasites/ μ l for *P. vivax* (the more pyrogenic of the two species – Price et al. 2007, Am J Trop Med Hyg, Ferreira et al. 2022, Lancet Reg Health), this approach increases our

chances of successful sequencing; cases of underperformance may also reflect other unidentified sample characteristics. We mention this hardly evitable underlying bias of passive collection in lines 472-474.

Fig. R13 Relationship between qPCR-based *P. vivax* parasitemia estimates and sequencing success.

For bioinformatic analysis and figures, all code should be made publicly available through github or similar (with link provided in the manuscript).

We have added analysis scripts and files to <https://github.com/fishntryps/pfpv> (line 704).

3. Analysis of transmission focuses on epidemiological and spatial covariates. What about drug-resistance markers? Are any of the clonal Pf groups associated with drug-resistance alleles?

None of the clonal *P. falciparum* groups from 2020-21 exhibit clear distinctions in known drug-resistance associated genetic variation relative to the rest of the sample set. We have assessed clone/cluster-related drug resistance marker profiles and selection signals at depth in a parallel study utilizing the same BioProject sequence data as well as large sample sets from 2016-17 and 2020. The present *P. falciparum* - *P. vivax* study therefore aimed not to interfere with that topic area. Also, without the longitudinal (2016-17 vs. 2020-21) component belonging to the partner study (which is cited at line 202), it is not very tractable to resolve possible resistance evolution dynamics in Guyana because cross-sectional spatial variation in resistance-related polymorphism appears very homogeneous / unpatterned within most of the Guiana Shield. Many alleles associated with resistance to previously used antimalarials (e.g., chloroquine, sulfadoxine, and pyrimethamine) are either fixed or near-fixed in the region (e.g., *crt* C72S + K76T, *mdr1* Y184F + N1042D + D1246Y, *dhfr* C50R N51I + S108N, *dhps* A437G + K540E + A581G, and *mdr2* I492V all occur in >98% of samples analyzed in the present study). An important exception to highly skewed resistance-associated allele frequencies represents *crt* C350R (ca. 54% frequency in Guyana between 2016 and 2021), a piperaquine resistance-associated mutation which shows regional temporal frequency changes and interacts epistatically with plasmepsin 2&3 gene copy number variation. The longitudinal partner study investigated whether the presence of *pfprt* C350R correlates to clonal temporal persistence or clonal size, but no significant relationship was observed. The longitudinal study also highlights a number of previously undescribed nonsynonymous mutations of interest, a few of which associate to particular clones, but their relevance is unknown. One example is a *kelch13* G718S mutation found in two Guyana samples representing clonal group # 57 in this

study.

MINOR COMMENTS:

1. Results: How much sequencing effort was applied and what Pv/Pf coverage was achieved for individual samples (BioSample IDs)? Could a supplementary table be added with this information? This will make the data more accessible to other researchers.

We have added mean depth of mapped reads to Supp. Table 2, lines 174-176, and Supp. Fig. 6.

2. Results: In the COI analysis, how were outliers processed? Was there any correlation with sequencing depth? THE REAL McCOIL can generate falsely elevated COI values in the setting of very low or very high coverage.

Please see response to Reviewer #3. We re-ran COI calling with equal SNP input sizes and filtering out sites with ≥ 200 mean or ≥ 150 sd read-depth, obtaining equal calls in 99.9% *P. falciparum* and 98.6% of *P. vivax* samples. This validation is also noted in Methods lines 589-591.

3. Results: The Pv clone recurrence findings are interesting, especially given that they were identified at HSPH in a blinded fashion and confirmed by the program in Guyana.

Yes! Given the scarcity of >0.90 IBD pairwise detections, the fact that many detections involved identical demographics (age, nationality, ethnicity, gender) clearly raised flags. Collaborators at the Ministry of Health could then answer a query on whether samples represented duplicates, with no additional patient information exposed to authors at HSPH.

4. Line 123: "Until 2019" implies that *P. vivax* was overrepresented until this date, when something changed. I think the authors mean to say "throughout the study period".

We amended to: "through 2019 (end of dataset)".

5. Fig 1: Before showing a map of Guyana, it would be nice to have a zoomed out version of its location within South America so readers who are not familiar with South American geography can understand its borders and overall location.

We added a zoomed out inset map to Fig. 1, top right.

6. Fig 1B: Why are people with Afroguyanese ethnicity more likely to have Pf than other groups? Because they are more likely to live in areas where Pf is more common? Are there other possible explanations?

Figure 1b shows the relative frequency of *P. falciparum* infections compared to *P. vivax* infections, and indeed indicates that the Afroguyanese ethnic categorization is more likely to harbor *P. falciparum* infections. The reason probably is twofold. First, people of African ancestry are known to be less susceptible to *P. vivax* infections as they rarely express the Duffy antigen receptor, which the *P. vivax* parasite uses to invade red blood cells. We now note this in lines 140 with a citation of Howes et al. 2015 (PLoS NTDs). As suggested by the reviewer, a second possibility is that the Afroguyanese are more likely to live or work in areas where *P. falciparum* is more common. However, this effect would likely be much more subtle given the extensive co-endemicity of the two parasite species. We therefore do not highlight this other possibility in the text.

7. Fig 1C caption (line 565): did the authors mean “x-axis are ordered by *P. vivax* count..” not y-axis?

Yes! Now amended, thanks very much.

8. Fig 2: Could major mining regions be highlighted on the map?

We have added an extra Supplementary Fig. 5 map showing principal cities/townships and 583 mining deposit coordinates available from the USGS Mineral Resources Data System (MRDS). According to the USGS, the MRDS is the best collection of reports it has available but also that updates are no longer occurring in systematic fashion. We therefore prefer these data to go in the supplement as opposed to Fig 2.

9. Lines 443-446: The concluding sentence implies that the disproportionate investment in Pf needs to be rectified. Given that the vast majority of global deaths are due to falciparum malaria, one could make a strong argument for continued “disproportionate” investment in Pf.

We have reworded this:

“It has been very important to prioritize investment in drugs, vaccines, and monoclonal antibodies for *P. falciparum* given its more frequent cause of acute severe disease, especially in children. Weaker focus on *P. vivax* control and specialized intervention however makes it likely that malaria will persist in many co-endemic regions after *P. falciparum* is eliminated, forestalling the ultimate goal of eliminating this major disease from all regions and demographics.”

10. The authors should provide a competing interests statement (it was not visible to me).

We added a statement (no competing interests – lines 706).

Reviewer #5 (Remarks to the Author):

REVIEWERS' COMMENTS

Reviewer #1 (Remarks to the Author):

My comments were adequately addressed in the revised manuscript.

Reviewer #2 (Remarks to the Author):

Reviewer #3 (Remarks to the Author):

My comments have been extensively addressed. I do not have any further comment.

Reviewer #4 (Remarks to the Author):

The authors have thoroughly addressed my comments.

Reviewer #5 (Remarks to the Author):
